### Zinc stimulation of phytoplankton in a low carbon dioxide, coastal Antarctic environment:

- 2 evidence for the Zn hypothesis
- Riss M. Kell<sup>1,+</sup>, Adam V. Subhas<sup>1</sup>, Nicole L. Schanke<sup>2</sup>, Lauren E. Lees<sup>3</sup>, Rebecca J. Chmiel<sup>1</sup>,
- Deepa Rao<sup>1</sup>, Margaret M. Brisbin<sup>1</sup>, Dawn M. Moran<sup>1</sup>, Matthew R. McIlvin<sup>1</sup>, Francesco
- Bolinesi<sup>4</sup>, Olga Mangoni<sup>4</sup>, Raffaella Casotti<sup>5</sup>, Cecilia Balestra<sup>6</sup>, Tristan J. Horner<sup>1</sup>, Robert B.
- Dunbar<sup>7</sup>, Andrew E. Allen<sup>8,9</sup>, Giacomo R. DiTullio<sup>2</sup>, Mak A. Saito<sup>1\*</sup>
- <sup>1</sup>Department of Marine Chemistry and Geochemistry, Woods Hole Oceanographic Institution,
- Woods Hole, MA, 02543, USA
- <sup>2</sup>Hollings Marine Laboratory, College of Charleston, Charleston, SC, 29424, USA
- <sup>3</sup>Department of Ecology and Evolutionary Biology, University of California at Irvine, Irvine,
- CA, 92697, USA

- <sup>4</sup>Department of Biology, Università degli Studi di Napoli Federico II, Complesso di Monte
- Sant'Angelo, Via Cinthia 21, 80126, Napoli, Italy
- <sup>5</sup>Department of Integrative Marine Ecology, Stazione Zoologica Anton Dohrn, Villa Comunale,
- 80121, Naples, Italy
- <sup>6</sup>National Institute of Oceanography and Applied Geophysics, 34010 Sgonico (TS), Italy
- <sup>7</sup>Doerr School of Sustainability, Stanford University, Stanford, CA, 94305, USA
- <sup>8</sup>Microbial & Environmental Genomics, J. Craig Venter Institute, San Diego, CA, 92093, USA
- 19 Scripps Institution of Oceanography, Integrative Oceanography Division, University of
- 20 California, San Diego, CA, 92037, USA
- 21 Correspondence to: Mak A. Saito (<u>msaito@whoi.edu</u>)
- <sup>+</sup>Formerly published under Riss Kellogg; now affiliated with Gloucester Marine Genomics
- 23 Institute, Gloucester MA, 01930

# Abstract.

| The ocean acts as a carbon sink, absorbing carbon from the atmosphere and resulting in                       |
|--------------------------------------------------------------------------------------------------------------|
| substantial uptake of anthropogenic CO <sub>2</sub> emissions. As biological processes in the oceans such as |
| net primary production (NPP) contribute significantly to this sink, understanding how they will              |
| shift in response to increasing atmospheric CO2 is necessary to project future ocean carbon                  |
| storage capacity. Macronutrient and micronutrient resource limitation within the oceans regulates            |
| NPP, and while some micronutrients such as zinc (Zn) are present at very low concentrations,                 |
| their ability to limit NPP has remained unclear. Zn is a key micronutrient used by phytoplankton             |
| for a multitude of metabolic functions, yet there have been few observations of its influence on             |
| natural oceanic phytoplankton populations. In this study, we observed Zn limitation of growth in             |
| the natural phytoplankton community of Terra Nova Bay, Antarctica, in addition to primary iron               |
| (Fe) limitation. Shipboard incubation experiments amended with Zn and Fe resulted in                         |
| significantly higher chlorophyll a content and dissolved inorganic carbon drawdown compared                  |
| to Fe addition alone. Zn and Fe stress response proteins detected in incubation and                          |
| environmental biomass provided independent verification of algal co-stress for these                         |
| micronutrients. We consider total biomass and low surface ocean pCO <sub>2</sub> as potential drivers of     |
| environmental Zn stress. This study definitively establishes that Zn limitation can occur in the             |
| modern oceans, opening up new possibility space in our understanding of nutrient regulation of               |
| NPP through geologic time, and we consider the future of oceanic Zn limitation in the face of                |
| climate change.                                                                                              |

# 1 Introduction

Primary productivity in the oceans is a key component of the global carbon cycle and is largely controlled by the availability of nitrogen (N), phosphorus (P), and iron (Fe). Yet there is increasing evidence that other micronutrients such as zinc (Zn), cobalt (Co), and vitamin B<sub>12</sub> can also influence phytoplankton productivity, often as secondary limiting nutrients after N, P, or Fe are added (Moore et al. 2013; Browning and Moore 2023). Zn can be particularly scarce in the photic zone (Bruland 1980; Jakuba et al. 2012) where total dissolved Zn (dZn<sub>T</sub>) can be below 0.2 nM in seawater due to biological uptake and complexation by organic ligands (Bruland 1989; Lohan et al. 2002; Baars and Croot 2011; Middag et al. 2019), which further lowers Zn bioavailability (Sunda and Huntsman 2000; Saito et al. 2008; Lhospice et al. 2017). Marine eukaryotic algae and copiotrophic bacteria possess a large metabolic demand for Zn that is on par with that of Fe (Sunda and Huntsman 2000; Mazzotta et al. 2021). Vertical profiles of dZn in the Southern Ocean have been measured previously. Zn has not historically been considered as a limiting micronutrient in the Southern Ocean due to the upwelling of nutrient-rich waters that bring dZn to nanomolar concentrations only a couple hundred meters below the surface. Yet nutrient-like profiles of dZn are evident throughout this region, with surface depletion due to biological uptake decreasing this large inventory in the upper water column (Fitzwater et al. 2000; Coale et al. 2005; Baars and Croot 2011; Sieber et al. 2020; Kell et al. 2024). .. Additionally, both model-based estimates (Roshan et al. 2018) and direct field measurements (Kell et al. 2024) of Zn uptake in this region have demonstrated a substantial biological demand for Zn in surface waters, leading to significant dZn drawdown.

This is consistent with and genomic and laboratory studies indicating an elevated Zn demand in

polar phytoplankton (Twining and Baines 2013; Ye et al. 2022).

Despite the scarcity of bioavailable Zn in the surface ocean and its high cellular demand, relatively few experimental studies have examined the ability of Zn addition to stimulate natural phytoplankton communities (Supplementary Table 1). These results have been variable with findings that include negative results (Scharek et al. 1997; Coale et al. 2003; Ellwood 2004), slight Zn stimulatory results (Crawford et al. 2003b), a "very small increase" relative to controls in an unreplicated experiment (Coale et al. 2003), Zn stimulation within Fe and Si uptake experiments (Franck et al. 2003), Zn primary and secondary limitation in the North Pacific in an unreplicated experiment (Jakuba et al. 2012), secondary Zn limitation after primary Si limitation in the Costa Rica Dome (Dreux Chappell et al. 2016), and enhanced Zn uptake rates under low pCO<sub>2</sub> (Xu et al. 2012). Whether due to the early negative results, the few positive findings, or the practical constraints of co-limitation studies in the field that limit the number of micronutrients that can be tested, it is our experience that there is currently no broad community recognition that zinc limitation is a process that could affect primary productivity in any region of the oceans, leaving the original 'zinc hypothesis' unresolved (Morel et al. 1994).

In contrast, laboratory studies have unequivocally demonstrated that marine phytoplankton can easily be Zn-limited in culture, and that Zn stress is exacerbated by low CO<sub>2</sub> due to an inability to synthesize the metalloenzyme carbonic anhydrase and resultant carbon colimitation (Morel et al. 1994; Buitenhuis et al. 2003; Sunda and Huntsman 2005). In this study, we reconcile these perspectives with a comprehensive, multipronged study of the natural phytoplankton assemblage in Terra Nova Bay (TNB), Antarctica, documenting evidence of Fe and Zn stress in a low pCO<sub>2</sub> coastal environment.

#### 2 Results

### 2.1 Biogeochemical characterization of Terra Nova Bay

Twenty-six stations within Terra Nova Bay (TNB) were temporally sampledover the course of one month (January 9 – February 18, 2018) during the 2017-2018 CICLOPS expedition (Fig. 1a; Supplementary Table 2) to concurrently characterize the natural progression of the phytoplankton bloom and biogeochemical changes in the water column (Kell et al. 2024). These stations were spatially distinct (each unique station was sampled once), but given that all stations were in relatively close proximity to each other within TNB (within a 52 km radius), we have combined all TNB station data to create a temporal analysis of the region. Surface waters within TNB had low (~200 µatm) seawater pCO<sub>2</sub> (Fig. 1b) which contrasted with measurements >400 µatm further from the study site (Fig. 1c). A large phytoplankton bloom was present as indicated by high (> 3000 ng L<sup>-1</sup>) chlorophyll fluorescence concentrations in January that waned into February (Fig. 1d). This observation of high productivity is characteristic of Antarctic polynya environments, which are recurring regions of open water surrounded by sea ice (Arrigo et al. 2012). This phytoplankton community initially consisted of a mixed assemblage of both diatoms as indicated by fucoxanthin (fuco, Fig. 1e) and the haptophyte *Phaeocystis* as verified by shipboard microscopy and as indicated by 19'hexanoyloxyfucoxanthin (19'-hex, **Fig. 1f**). Surface fucoxanthin concentrations >200 ng L<sup>-1</sup> were observed at the late TNB stations (Fig. 1e) while 19'-hex decreased to ~20 ng L<sup>-1</sup> (Fig. 1f), indicating that the stations sampled in late February were dominated by diatoms rather than Phaeocystis. This was consistent with historical observations of phytoplankton succession patterns in TNB (DiTullio and Smith 1996; Smith et al. 2006; Mangoni et al. 2019). Additionally, we observed pronounced depletion of total dissolved Zn in surface waters across all TNB stations, with an average concentration of  $0.82 \pm 0.47$  nM at 10 m (Fig. 1g). Notably, as

the bloom progressed, this depletion extended progressively deeper into the water column (Fig. 1g), indicative of strong Zn uptake and export from the euphotic zone.

Total Zn uptake ( $\rho$ Zn, measured concurrently using a stable isotope tracer method) (Kell et al. 2024) was highest in the shallow euphotic zone in early January and waned into February (Fig. 1h), following trends seen in chlorophyll fluorescence (Fig. 1d) and 19'-hex (Fig. 1f). This  $\rho$ Zn trend was consistent with laboratory studies demonstrating the substantial Zn requirements of both diatoms and *Phaeocystis antarctica* (Saito and Goepfert 2008; Kellogg et al. 2020). Across all TNB stations, total dissolved Fe (dFe<sub>T</sub>) in the upper 50 m remained below 1 nM (Fig. 1i) as observed previously in this region (Fitzwater et al. 2000). In the Ross Sea, dissolved iron (dFe) has previously been demonstrated to be the primary limiting nutrient for phytoplankton growth (Martin et al. 1990; Coale et al. 2003; Sedwick et al. 2011).

### 2.2 Biogeochemical characterization of the incubation study site

Within TNB station, station 27 (referred to as the "experimental site" herein) was chosen for the multifactor shipboard incubation experiment (**Fig. 1a,b**; red star). This site harbored a coastal bloom and was biologically and chemically characterized as having high *in situ* chlorophyll *a* levels (maximum of 3259 ng L<sup>-1</sup> at 30 m; **Fig. 1j**) and was comprised of diatoms as indicated by fucoxanthin and *Phaeocystis* as indicated by 19'-hex (**Fig. 1k**). A decrease in surface total dissolved inorganic carbon (DIC<sub>T</sub>; 2181 μmol kg<sup>-1</sup> at 15 m compared to the deep water (200-1065 m) average of 2224 ± 2.1 μmol kg<sup>-1</sup>, **Fig. 1l**) was also observed. Within the water column, dZn demonstrated a pronounced decrease from 5.1 nM at 50 m to 0.9 nM at 10 m, representing an 82% decrease (and a 76% decrease comparing the minimum dZn<sub>T</sub> value at 10 m

to the average deepwater (210 – 1000 m) concentration of 3.9 nM  $\pm$  0.4; **Fig. 1m**), consistent with prior observations of surface dZn depletion in this region (Fitzwater et al. 2000).

Observations of rapid Zn uptake (46 pmol L<sup>-1</sup> d<sup>-1</sup> at 10 m) at the experimental site (**Fig. 1n**) likely contributed to this surface depletion, as Zn uptake rates of this magnitude are of the appropriate scale to induce the multi-nanomolar surface water depletion during the austral spring and summer season (Kell et al. 2024). Consistent with high macronutrient abundance in this region, surface macronutrient concentrations were partially depleted at the experimental site with 64%, 46%, and 29% decreases in nitrate+nitrite (N+N), phosphate (P), and silicate (Si), respectively, comparing 10 m and average deep water (200 – 1000 m) values (**Fig. 10**).

# 2.3 Evidence for Zn stimulation of phytoplankton: experimental site shipboard incubations

A multifactor incubation experiment was conducted using surface waters collected at the experimental site by trace metal clean fish sampler (7 m) fed into a shipboard cleanroom to examine controls on net primary productivity, with triplicate treatments of Zn amended (+Zn; 2 nM as ZnCl<sub>2</sub>), Fe amended (+Fe; 1 nM as FeCl<sub>2</sub>), and Fe and Zn amended (+Fe+Zn) incubations, in addition to unamended controls. Addition of Fe alone (+Fe) resulted in significantly higher chl a content compared to controls (p = 9.5e-5) after six days (T6) (**Fig. 2a**), demonstrating primary Fe limitation as observed previously in the Ross Sea (Martin et al. 1990; Mangoni et al. 2019). However, addition of Zn alone (+Zn) also resulted in significantly higher chl a content compared to the controls (p = 0.011), implying that a subset of the incubated phytoplankton population benefitted from the addition of Zn alone, without additional Fe (**Fig. 2a**). This observation is consistent with independent co-limitation (Saito et al. 2008), where two nutrients (such as Fe and Zn) each independently limit different subpopulations or processes, and adding

either nutrient alone yields a response. The combined addition of +Fe+Zn resulted in the highest average chl a content among all treatments at T6, with  $4.5 \pm 0.21~\mu g~L^{-1}$  compared to  $3.9 \pm 0.35~\mu g~L^{-1}$  and  $2.8 \pm 0.15~\mu g~L^{-1}$  achieved by +Fe and +Zn alone, respectively, demonstrating additive co-limitation between Zn and Fe (Sperfeld et al. 2016). The Zn stock solution was analyzed to confirm these results were not caused by inadvertent Fe contamination (see Methods). Significant differences in seawater chemistry were also observed within these incubations over time, with larger decreases in DIC<sub>T</sub> in all metal treatments compared to the control (-12.7  $\mu$ mol kg<sup>-1</sup> for +Fe (p = 5.3e-6), -8.2  $\mu$ mol kg<sup>-1</sup> for +Zn (p = 5e-5), and -18.5  $\mu$ mol kg<sup>-1</sup> for +Fe+Zn (p = 2.2e-16); **Fig. 2b**). The decrease in DIC<sub>T</sub> observed with +Fe+Zn was significantly larger than that achieved with +Fe alone (p = 4.4e-3; **Fig. 2b**). Statistically significant differences in measured parameters among treatments are summarized in **Supplementary Table 3.** 

Further consistent with the observed Zn stimulation of biomass in the incubations, the largest decreases in macronutrient (P and N+N) concentrations in these incubations at T6 were observed in the +Fe+Zn treatment (**Supplementary Figure 1a,b**), as was the largest increase in particulate organic carbon (POC; **Supplementary Figure 1c**). POC collected from the +Zn and +Fe+Zn incubations was characterized by larger C:N atomic ratios (5.9 and 6.2, respectively) compared to the +Fe and T6 control (5.2 and 5.3; **Supplementary Figure 1d**). Significantly higher bacterial abundances in both +Fe (p = 9.1e-4) and +Fe+Zn (p = 6.3e-4) treatments relative to the T6 control (**Supplementary Figure 1e**) indicated the alleviation of bacterial Fe limitation, consistent with prior reports (Obernosterer et al. 2015; Fourquez et al. 2020; Sun et al. 2021).

At the conclusion of the incubation experiments, biomass was collected by serial filtration through 5 mm and 0.2 mm filters, and the 0.2-5  $\mu$ m fraction was extracted for

proteomic analysis (see Methods) and analyzed for biomarkers of Zn and Fe stress. We detected both algal Fe- and Zn-stress proteins, which provided an independent line of evidence corroborating the results described above (Fig. 2c). This included the detection of the Zn/Co responsive protein ZCRP-A (a putative Zn chaperone) (Kellogg et al. 2022a) as a biomarker of Zn stress as well as the iron starvation-induced proteins ISIP1A, ISIP2A and ISIP3 (ISIPs) as biomarkers of Fe stress (Supplementary Table 4). The ISIPs represent a group of unrelated proteins that are upregulated under Fe limitation in various algal species. ISIP1 proteins are responsible for endocytosis of siderophore-bound iron, ISIP2 proteins are involved in Fe<sup>3+</sup> uptake, and ISIP3 has been suggested to act as an Fe storage protein (Allen et al. 2008; Behnke and LaRoche 2020). RUBISCO abundance within each treatment is shown in Fig. 2c as a proxy for the potential phytoplankton production. Within the T6 incubation biomass, there was an increased abundance of ISIPs in the control and +Zn treatment, and a decrease in ISIP protein abundance within the +Fe and +Fe+Zn treatments, consistent with primary Fe limitation and the expected response to Fe addition (Fig. 2c). ISIPs were taxonomically assigned to diatoms, *Phaeocystis*, and dinoflagellates (**Fig. 2d**). The strongest expression of ZCRP-A protein was detected in the +Fe treatment (Fig. 2c,d) indicative of Fe addition driving the community towards increased Zn stress. Notably, ZCRP-A was still detected in the +Fe+Zn treatment (Fig. 2c,d), implying that the added Zn was unable to completely satiate Zn demand as phytoplankton biomass increased (as indicated by the increase in chl a at T6, Fig. 2a), despite added Zn (2 nM) being double that of added Fe (1 nM). Sequence analysis of the contigs identified as ZCRP-A homologs in these incubations revealed that all contigs contained one or more canonical conserved motifs found in COG0523 family proteins such as ZCRP-A (Supplementary Figure 2). Coupled with evidence from prior laboratory studies (Kellogg et al. 2022a), this provides

further support for the role of ZCRP-A in responding to Zn scarcity. ZCRP-A proteins were taxonomically assigned to chlorophytes, dinoflagellates, and *Phaeocystis*, with the detection of *Phaeocystis* ZCRP-A only in the +Fe treatment (**Fig. 2d**). The detection of ZCRP-A attributed to *Phaeocystis*, but the nondetection of ZCRP-A attributed to diatoms implies that either ample diatom biomass was not captured on the analyzed filters due to being filtered out by the >5mm pre-filter, or that diatoms present in these incubations (as indicated by diatom RUBISCO; **Fig. 2d**) were outcompeting *Phaeocystis* for Zn. Our observations of Fe and Zn biomarkers shifting in abundance in response to their respective metal treatment provides independent evidence for Zn/Fe co-limitation.

#### 2.4 Taxonomic characterization of incubation results

To characterize the phytoplankton species responding to metal amendment, we measured phytoplankton pigments within the shipboard incubations over time, which revealed a diverse taxonomic response to metal amendments. Measured pigments included fucoxanthin (fuco), 19'-hexanoyloxyfucoxanthin (19'-hex), prasinoxanthin (prasino), chlorophyll b (chl b), and chlorophyll c3 (chl c3). Fuco is produced by both diatoms and by *Phaeocystis* under certain conditions, while 19'-hex and chl c3 are indicative of *Phaeocystis* in the Southern Ocean (DiTullio et al. 2007). Fuco:19'-hex ratios significantly increased in the +Fe (p = 4.2e-4) and +Fe+Zn treatments (p = 2.7e-3) (**Supplementary Figure 3a**) due to no significant change in fuco (**Supplementary Figure 3b**) and decreased 19'-hex (**Supplementary Figure 3c**) relative to the T6 control. *Phaeocystis* contributions to total fuco concentrations are typically minimal at the low Fe levels of the Ross Sea, though *Phaeocystis* can revert to making fuco rather than 19'-hex when released from Fe limitation (DiTullio et al. 2007), as was evident in these incubations by

decreased 19'-hex:chl c3 ratios within the +Fe and +Fe+Zn treatments (Supplementary Figure 3d). Phaeocystis therefore likely contributed to total fuco by responding to Fe addition. Notably, significant decreases in both fuco:chl a and 19'-hex:chl a (Supplementary Figure 3e,f) in all treatments compared to the T6 control indicated that other phytoplankton groups contributed to chl a (Fig. 2a) without contributing fuco nor 19'hex. Increases in chl b (Supplementary Figure 3g) and prasinoxanthin (Supplementary Figure 3h) suggest that small green algae such as chlorophytes and prasinophytes also responded to +Fe and +Zn independently, consistent with the detection of chlorophyte ZCRP-A in these incubations (Fig. 2d). Photosynthetic efficiency of photosystem II (Fv/Fm) significantly increased with +Fe (p = 0.011) and with +Fe+Zn (p =0.0036) at T4 (day 4) compared to T4 controls, but did not significantly increase with +Zn alone, implying Fv/Fm may not be useful as a diagnostic for Zn stress and that caution should be used in interpreting its signals universally (Supplementary Figure 3i). No significant difference in Fv/Fm was observed among treatments at T6. Selective zooplankton grazing on small diatoms and solitary *Phaeocystis* cells may have played a role in affecting phytoplankton biomass and the observed pigment:chl a ratios. For instance, higher ratios of phaeophytin:total phaeopigments were observed in +Fe and +Zn amended incubations (Supplementary Figure 4) which may reflect grazing on solitary *Phaeocystis* cells, as high phaeophytin:total phaeopigments ratios were previously observed in *Phaeocystis* dominated waters of the Ross Sea (DiTullio and Smith 1996).

#### 2.5 Detection of Zn- and Fe-stress protein biomarkers in the water column

Metaproteomic and metatranscriptomic analyses of biomass within the water column at the experimental site provided additional confirmation of the incubation results, as we detected Zn- and Fe- stress-response proteins present within the water column, which were therefore naturally present without influence from incubation conditions. In addition to ISIPs and ZCRP-A, we detected ZCRP-B (a putative membrane-tethered Zn-binding protein) (Kellogg et al. 2022a), Zrt/Irt-like (ZIP) transporters (which are known to be used by marine phytoplankton for uptake of Zn<sup>2+</sup> and other divalent metal cations (Allen et al. 2008; Milner et al. 2013; Bender et al. 2018)), and  $\theta$  (theta) and  $\delta$  (delta) carbonic anhydrases (CAs).  $\theta$ -CAs with Zn<sup>2+</sup> coordination sites have been documented in diatoms (Jensen et al. 2020a), including the polar diatom *Chaetoceros neogracile* RS19 (Kellogg et al. 2022b), but no studies to date have investigated enzyme activity nor efficiency with Co<sup>2+</sup> or Cd<sup>2+</sup>. In contrast,  $\delta$ -CA (i.e, *Thalassiosira weissflogii* TWCA1) is known to function with either Co<sup>2+</sup> or Zn<sup>2+</sup> as a cofactor (Lane and Morel 2000b) conferring metabolic flexibility when Zn<sup>2+</sup> is scarce.

Both proteins and transcripts of Zn and Fe stress biomarkers (ZCRP-A and ISIPs) were observed throughout the water column at the experimental site. RUBISCO, ZCRP-A, and ISIP protein spectral counts were most abundant at the surface and decreased with depth within the within the 3µm size fraction (Fig. 3a-c), consistent with the depletion of trace metals in the photic zone due to high-biomass bloom conditions. ZCRP-A was detected in both 3 and 51 µm filter pore-size fractions (Fig. 3b) and was predominantly attributed to *Phaeocystis* and the diatom genus *Chaetoceros* in the euphotic zone, and predominantly to *Phaeocystis* and the diatom genus *Pseudo-nitzschia* in the mesopelagic zone (Fig. 3i, Supplementary Figure 5a). The presence of *Phaeocystis* below the photic zone is consistent with prior observations of rapid export of *Phaeocystis* cells (DiTullio et al. 2000). Throughout the water column, ISIPs were predominantly attributed to *Phaeocystis* and to the diatom genera *Fragilariopsis*, *Chaetoceros*, and *Pseudo-nitzschia* (Fig. 3i, Supplementary Figure 5b). We note that ZCRP-B, a protein also found to be upregulated in marine diatoms under low Zn/Co and characterized as a putative

membrane-tethered Zn/Co protein ligand (Kellogg et al. 2022a) was most abundant in the 0.2 μm fraction throughout the water column (**Fig. 3d**). As ZCRP-B shares ~30% similarity to the bacterial ABC-type nickel transporter component NikA, spectral counts within the bacterial 0.2 μm fraction most likely reflect true bacterial NikA. BLAST analysis of all ZCRP-B contigs confirmed that all ZCRP-B hits across all size fractions corresponded to bacteria (**Fig. 3i**).

The assignment of the majority of ZCRP-A and ISIP proteins to *Phaeocystis* in the upper water column provides additional evidence that *Phaeocystis* was likely Zn/Fe co-limited at the study site, consistent with incubation results (**Fig. 2d**).

ZCRP-A belongs to the phylogenetically complex COG0523 family, with some family members showing functional divergence (that is, activity using different metal cofactors) among paralogs (Blaby-Haas and Merchant 2012; Edmonds et al. 2021). Here, we infer a Zn-responsive function for the identified ZCRP-A contigs based on their homology to *T. pseudonana* and *P. tricornutum* ZCRP-A proteins, which we have previously characterized as Zn-responsive (Kellogg et al. 2022a). To further support this inference, we used SHOOT (Emms and Kelly 2022) to place each ZCRP-A contig within a phylogenetic context. Of the 21 unique contigs assigned as ZCRP-A homologs, 19 were confirmed to be *T. pseudonana* orthologs, while 2 were assigned as orthologs to the Zn-related COG0523 *E. coli* proteins YjiA and YeiR, implying a minor prokaryotic source (Supplementary Table 5). The placement of the majority of these contigs within diatom clades supports our interpretation that these homologs are Zn-related.

ZIP proteins were almost solely detected in the 51 μm fraction, likely due to the capture of abundant *Phaeocystis* colonies and chain-forming diatoms (**Fig. 3e; Supplementary Figure 5c**). ZIP family transporters are functionally diverse and capable of transporting multiple divalent metal cations, including both Zn<sup>2+</sup> and Fe<sup>2+</sup> (Blaby-Haas and Merchant 2012), with

diatom homologs of ZIP1 known to be upregulated under Fe stress (Lampe et al. 2018). Given the co-limitation of Fe and Zn at the study site, it is difficult to determine which metal these ZIP transporters were primarily mediating.

The increased abundance of diatom  $\theta$ -CA and  $\delta$ -CA proteins within the water column (**Fig. 3f,g**), as well as transcripts for the diatom Cd carbonic anhydrase CDCA, which can replace Zn<sup>2+</sup> with Cd<sup>2+</sup> as the catalytic cofactor (Lane and Morel 2000a), in the 3  $\mu$ m and 51  $\mu$ m fractions at 200 m (**Fig. 3h**) was indicative of a sinking, prior diatom bloom event (Subhas et al. 2019).  $\theta$ -CA and  $\delta$ -CA were predominantly taxonomically assigned to the diatom genera *Chaetoceros* and *Pseudo-nitzschia*, respectively, while CDCA transcripts belonged to the diatom genera *Chaetoceros* and *Corethron* (**Fig. 3i**). The presence of  $\theta$ -CA, but lack of  $\delta$ -CA, assigned to *Chaetoceros* is consistent with proteomic analysis of the polar diatom *Chaetoceros neogracile* RS19 grown in culture under Zn limiting conditions (Kellogg et al. 2022b).

### 2.6 Zn:P ratios of the surface seawater at the experimental site

A third independent line of evidence for the nutritional influence of Zn scarcity on TNB phytoplankton was obtained from *in situ* cellular stoichiometry. Particulate Zn:P ratios (Zn:P) analyzed from biomass collected at the surface of this experimental station were consistent with ratios from Zn-limited culture studies. Particulate Zn:C ratios reported previously in Zn-limiting culture studies of the diatom *Thalassiosira pseudonana* (Sunda and Huntsman 2005) were converted to Zn:P ratios using the Redfield ratio (Redfield 1958) (**Supplementary Table 6**). We then compared these ratios and associated growth rates with particulate Zn:P measured within biomass collected at 10, 25, 50 and 100 m at the experimental site. At each of these surface depths, Zn:P measured at the experimental site was ~ 2E-4 mol:mol, which, in comparison to

cultured diatom Zn:P ratios, fell within the range of severely Zn-limited growth rates

(Supplementary Figure 6), again demonstrating the propensity for Zn-limited growth in this region and corroborating the incubation results.

#### 3 Discussion

Antarctic waters are generally considered to not be prone to Zn limitation, given that high (> 1nM) dZn concentrations are typically observed in surface Southern Ocean waters (Coale et al. 2003). However, we observed multiple independent lines of evidence from both the field incubation experiment (chlorophyll, DIC, Zn and Fe biomarker proteins) and contextual environmental biogeochemical data of the water column at the incubation site (dZn, Zn uptake rates, pigments, cellular Zn:P stoichiometry, metaproteomic, and metatranscriptomic analyses) demonstrating that phytoplankton within Terra Nova Bay of the Ross Sea, Antarctica, were experiencing Zn and Fe nutritional stress.

Multiple factors could be considered as potential drivers in the creation of Zn-limiting conditions in the field, including Zn demand imposed by total biomass and the species comprising this biomass. The phytoplankton bloom observed during this expedition was comprised primarily of diatoms and *Phaeocystis*, consistent with previous Ross Sea seasonal blooms (Smith et al. 2006; Arrigo et al. 2012; Mangoni et al. 2019), and which contributed to the observed high Zn uptake rates and thus surface Zn depletion (Kell et al. 2024), resulting in nutrient-like dZn profiles throughout TNB.

Our field observation of Zn limitation was made in an environment characterized by diminished pCO<sub>2</sub>, which we consider as a factor potentially driving Zn stress. We observed a substantial drawdown of surface seawater pCO<sub>2</sub> to 221 µatm at the incubation site (a ~45% decrease compared to offshore waters in the Ross Sea measured during the same time frame; **Fig.** 

**1b**). Biology was the driver of this decrease in pCO<sub>2</sub>, rather than freshwater input from glacial and sea ice melt. This is evident in the physicochemical data, where over the measured salinity range (S=33.6-34.8), the effect of simple dilution by freshwater input (DIC=Total Alkalinity=0) would result in a reduction of pCO<sub>2</sub> by only ~8-9 ppm. The signals we observe are much larger than that, consistent with a large phytoplankton uptake driver. The total alkalinity (TA) also does not change proportionally with DIC in this region, which is also not consistent with dilution driving a conservative mixing of TA and DIC.

Laboratory studies have unequivocally demonstrated that marine phytoplankton can easily be Zn-limited in culture due to their large Zn requirement, and that this effect is exacerbated at low pCO<sub>2</sub> (Morel et al. 1994; Sunda and Huntsman 2005) due to the use of Zn as a required catalytic cofactor within carbonic anhydrase (CA) metalloenzymes (Sunda and Huntsman 2005). CAs catalyze the reversible dehydration of HCO<sub>3</sub><sup>-</sup> to CO<sub>2</sub>, the substrate required by the carbon fixing enzyme RUBISCO. As HCO<sub>3</sub><sup>-</sup> constitutes about 90% of the dissolved inorganic carbon (DIC) pool in the surface ocean, sufficient CA activity prevents carbon stress in marine phytoplankton by ensuring adequate CO<sub>2</sub> supply to RUBISCO. It has therefore been hypothesized that the combination of high biomass and resulting low CO<sub>2</sub> may cause severe Zn depletion that may limit algal growth rates due to lack of Zn and thus reduced CA activity, and thus reduced availability of carbon for photosynthesis (Morel et al. 1994; Sunda and Huntsman 2005). This Zn-C limitation relationship is referred to as 'biochemically dependent co-limitation', in which the availability of one nutrient is essential for the acquisition or utilization of another nutrient, especially at low concentrations (Saito et al. 2008).

To explore this in the context of our field observations, using the available quantitative constraints on Zn and CO<sub>2</sub> co-limitation thresholds available from the literature (see Methods),

we estimated that the threshold for Zn-CO<sub>2</sub> limitation in culture synthesized across many alga occurs at 259 µatm pCO<sub>2</sub>. We then compared this laboratory-determined Zn/C limitation threshold estimate to both the *in situ* 221 µatm pCO<sub>2</sub> measured at our field study site, and to the historical, global trend in surface ocean pCO<sub>2</sub> (Fig. 4a,b). Global surface ocean pCO<sub>2</sub> levels are rapidly rising above both the laboratory-estimated 259 µatm pCO<sub>2</sub> Zn/C limitation threshold and our field observation value of 221 µatm (Jiang et al. 2023) (Fig. 4a,b). Though only a fraction of the modern-day surface ocean is currently at  $\leq 250$  ppm pCO<sub>2</sub> (predominantly comprised of polar regions; Fig. 4c), this represents a large decrease in oceanic extent compared to only 100 years ago (Fig. 4d). Even though this may move the majority of oceanic regions farther from Zn and C limitation thresholds, there continue to be highly productive and episodic coastal blooming events that induce significant pCO<sub>2</sub> drawdown (Harrison et al. 2018; Dai et al. 2022). These coastal regions are increasingly recognized as being disproportionally significant contributors to global ocean carbon export (with respect to their area), particularly at the high latitudes (Harrison et al. 2018; Dai et al. 2022), and will hence continue to be prone to Zn stress at low CO<sub>2</sub> as we have observed. Many other coastal regions have been observed to experience depressed CO<sub>2</sub> such as the Amundsen Sea (Tortell et al. 2012), Amazon River plume (Valerio et al. 2021), the west Florida Shelf (Robbins et al. 2018), the East China Sea (Shim et al. 2007), the Northern Gotland Sea (Schneider and Müller 2018), and Monterey Bay, California (Chavez et al. 2018) to name a few examples. On the other hand, it is likely that despite rising pCO<sub>2</sub> levels, some coastal regions will continue to experience episodic or persistent low pCO<sub>2</sub> due to high productivity (as observed in this study), freshwater inputs, or other regional processes. Though we do not attempt to model future pCO<sub>2</sub> dynamics in these areas, our results suggest that Zn status may continue to be an important physiological constraint under low pCO<sub>2</sub> conditions, particularly in productive

coastal systems. As such, Zn limitation should be considered as part of the broader framework for understanding carbon cycling in these regions, especially as they play a disproportionate role in global carbon export.

#### **4 Conclusions**

Given the great challenge of conducting Zn manipulation experiments without contamination, we did not try to manipulate pCO<sub>2</sub> as an additional experimental treatment. Instead, we actively sought out a low pCO<sub>2</sub> environment for the study site, building on prior laboratory studies and a cadmium (Cd) pCO<sub>2</sub> field study (Cullen et al. 1999). The interaction of Zn (and Cd and Co) with CO<sub>2</sub> is an important area of future research, particularly in coastal environments. With the continuing rise in atmospheric and surface ocean pCO<sub>2</sub> levels, broader changes in the biogeochemical cycling of Zn and other bioavailable trace metals will likely occur within the oceans, influencing NPP and thus total ocean carbon storage capacity. These low pCO<sub>2</sub> conditions environments that routinely occur in numerous coastal environments globally should be further examined for Zn effects in addition to carbon uptake dynamics in different temperature environments (Tortell et al. 2008; Dai et al. 2022). While there are elaborate biochemical capabilities available to many marine algae for dealing with Zn scarcity (Kellogg et al. 2022a), our results suggest that the geographic extent of possible Zn/C co-limiting environments may further decrease in the coming decades with rising anthropogenic CO<sub>2</sub> emissions. Despite this, the biochemical demand for Zn in marine organisms remains substantial, with cellular demand rivaling that of Fe. The multitude of metabolic functions requiring Zn, including but not limited to carbonic anhydrase activity, implies the need for further exploration of Zn influences on primary productivity in a changing ocean environment.

#### 5 Materials and Methods

### 5.1 Study area and sample collection

Sample collection occurred during the CICLOPS (Cobalamin and Iron Co-limitation of Phytoplankton Species) expedition (NBP18-01) aboard the RVIB *Nathaniel B. Palmer*,

December 11, 2017 – March 3, 2018 in the Amundsen Sea and Ross Sea of the Southern Ocean (Fig. 1a). Station metadata is provided in Supplementary Table 2. All stations were assumed to be representative of TNB during this temporal study (as evident in the total dissolved metal, macronutrient, and chlorophyll *a* datasets). Water samples for dissolved trace metal analyses were collected using trace metal sampling protocols described previously (Cutter and Bruland 2012). A trace metal clean rosette suspended on a Kevlar line and equipped with twelve 8L X-Niskin bottles (Ocean Test Equipment) was used to collect seawater at depths ranging from 10 – 600 m. Niskin bottles were transported to a positive-pressure trace metal clean shipboard van for filtration upon surfacing. Total fluorescence on the vertical profiles was measured using an ECO chlorophyll fluorometer (Wet Labs) equipped to the rosette. The rosette also included instrumentation for measuring conductivity and temperature (Sea-Bird Electronics).

# **5.2 Preparation of plasticware**

Polyethylene and polycarbonate sampling and incubation bottles were rigorously cleaned to remove trace metal contaminants before use. Bottles were rinsed with 18.2  $\Omega$  Milli-Q water (Millipore), soaked for 72h in <1% Citranox detergent, rotated, soaked for an additional 72h, and then rinsed five times with Milli-Q water. Bottles were then filled with 10% HCl (Baker instaranalyzed) by volume and soaked for a minimum of one week, rotated, and soaked for another

week. Bottles were then rinsed five times with dilute acid (HCl, pH 2) and stored double-bagged in plastic zip bags. All cleaning work was conducted in a Class 100 clean room.

# 5.3 Underway seawater pCO<sub>2</sub> partial pressure

Surface water pCO<sub>2</sub> measurements were conducted aboard the RVIB *Nathaniel B*.

Palmer using an underway method consisting of an air-water equilibrator and IR CO<sub>2</sub> analyzer developed and operated by the Lamont-Doherty Earth Observatory (LDEO) group (Takahashi et al. 2020). A complete data report and sensor list are available: <a href="https://service.rvdata.us/data/cruise/NBP1801/doc/NBP1801DATA.pdf">https://service.rvdata.us/data/cruise/NBP1801/doc/NBP1801DATA.pdf</a> (last access: 14 December 2024).

#### **5.4 TDIC and POC measurements**

Total alkalinity (TA) and dissolved inorganic carbon (DIC) were measured on CTD and incubations samples in near real-time aboard the NBP. Dissolved inorganic carbon (DIC) and total alkalinity (TA) samples were collected following previously established protocols (Dickson et al. 2007). DIC analyses were conducted within ~4 h of collection. We acidified 1.25 mL of sample using an automated custom-built injection and bubble stripping system coupled to an infrared gas analyzer (LICOR LI7000) and integrated the infrared absorption signal versus time for each stripped gas sample to yield a total mass of CO<sub>2</sub>. Each sample was analyzed in triplicate or greater. Since microbubbles regularly formed as samples warmed between sample acquisition and DIC analysis, every integration curve was visually inspected and those curves that exhibited evidence for bubbles were rejected. Certified reference materials (Dickson CRM batch 169) were analyzed between every 3 to 4 unknowns. The estimated precision based upon unknowns (>860

samples run in triplicate) and CRM replicates (n = 738) was  $\pm$  2.0  $\mu$ mol kg<sup>-1</sup> ( $\pm$ 1 SD). Analyses for TA on filtered samples were completed within ~12 h of collection by using a potentiometric titrator (Metrohm 855 Robotic Titrosampler) (DeJong et al. 2015). The estimated precision based on replicate analyses of CRMs (n = 195) was  $\pm$  2.6  $\mu$ mol kg<sup>-1</sup> ( $\pm$ 1 SD).

# 5.5 Analysis of historical atmospheric and surface ocean pCO<sub>2</sub> trends

Decadal surface ocean pCO<sub>2</sub> reconstructions (Jiang et al. 2023) were downloaded, binned by decade, and plotted using the 'violinplot' in MATLAB. Atmospheric pCO<sub>2</sub> data was assembled from the running Mauna Loa record (Keeling et al. 1976), and from measurements made on Antarctic firn ice (Etheridge et al. 1996).

# 5.6 Calculation of Zn-and pCO<sub>2</sub> co-limitation of phytoplankton thresholds

There are few experimental measurements of Zn- and pCO<sub>2</sub>-co-limitation, either in the lab or *in situ*. This study documented Zn of a natural phytoplankton assemblage in the field at a pCO<sub>2</sub> of ~220 ppm. In the literature, several models exist to interpret co-limitation (Buitenhuis et al. 2003; Saito et al. 2008). For this study we chose to use the biochemically dependent co-limitation model for growth rate ( $\mu$ ):

$$\mu = V_{max} \frac{\left[CO_{2(aq)}\right]}{\varphi K_s + \left[CO_{2(aq)}\right]}, \quad (1)$$

where  $V_{max}$  is the maximum growth rate,  $[CO_{2(aq)}]$  is the aqueous  $CO_2$  concentration of the growth medium in micromoles per kilogram of seawater,  $K_s$  is the half-saturation constant in micromoles per kilogram of seawater, and  $\varphi$  is a Zn-dependent growth term:

$$\varphi = \frac{[dZn] + K_{s,Zn}}{[dZn]} \tag{2}$$

Here, the dissolved Zn concentration in the growth medium [dZn] is modified by a Zn-dependent saturation constant  $(K_{s,Zn})$ . Few studies have enough experimental data to robustly establish a kinetic relationship between [dZn] and  $[CO_{2(aq)}]$ , so we compiled several estimates for these terms from the literature. Diatom growth rates under pCO<sub>2</sub> limitation were taken from Riebesell et al. (1993) (Riebesell et al. 1993). Reported pH and temperature measurements for each treatment, a total alkalinity of 2300 µmol kg<sup>-1</sup>, and a salinity of 35 were used to calculate aqueous CO<sub>2</sub> concentrations using CO2SYSv3.2.1.(Sharp et al. 2023). Reported  $V_{max}$  and  $K_s$  for D. brightwellii, T. punctigera, and R. alata were 1.46, 1.30, and 0.93 d<sup>-1</sup>, and 1.4, 1.2, and 2.1  $\mu$ mol kg<sup>-1</sup>, respectively. Values for coccolithophore growth ( $K_s = 0.97$  μmol kg<sup>-1</sup>,  $V_{max} = 4.7$  d<sup>-1</sup>) were taken from Krumhardt et al. (2017) (Krumhardt et al. 2017). A value for  $K_{s,Zn}$  of 300 pmol L<sup>-1</sup> was taken from Buitenhuis et al. (2003). This value is for the coccolithophore *E. huxleyii* generated under varying CO<sub>2</sub> and Zn conditions, and Zn response growth curves under single CO<sub>2</sub> conditions (ambient) are similar to other diatoms like *T. pseudonana* (Sunda and Huntsman 1995). The value of 300 pmol L<sup>-1</sup> appears high, but is tied to the functional form of the biochemically co-limitation equation (Buitenhuis et al. 2003; Saito et al. 2008). Based on the same dataset and different models for co-limitation, Buitenhuis et al. (2003) arrived at  $K_{s,Zn}$ values of ranging from 38 pmol L<sup>-1</sup> to 300 pmol L<sup>-1</sup>. They calculated Zn-limitation alone, at CO<sub>2</sub>replete conditions, of 19 pmol  $L^{-1}$ . Thus, the chosen value of  $K_{s,Zn}$  is not a reflection of high Zn demand but determined by the functionality of biochemical co-limitation by Zn and C. To calculate φ, a surface ocean Zn concentration of 50 pmol L<sup>-1</sup> was assumed (Bruland 1980; Wyatt et al. 2014). While these concentrations reflect total dissolved Zn, the relationship

between bioavailable free Zn and dZn, especially in the field, remains unclear. Eq. 1 was then used to calculate effective  $CO_2$  concentrations (and thus  $pCO_2$  values) at which growth is halved, or in other words,  $\mu = 0.5 V_{max}$ . We note that this calculation is distinct from the  $CO_2$  half-saturation constant because of the co-limitation by Zn. The median  $pCO_2$  threshold for 50% growth from the three diatom species was 278 ppm. Including coccolithophores decreases the median to 259 ppm. These values are slightly higher than the *in situ* evidence for Zn limitation at 220 ppm presented in the present study. Our results cannot be considered as an upper bound for Zn- $CO_2$  limitation, but serve as evidence for growth limitation under those specific environmental conditions.

### 5.7 Analyses of total dissolved metals using isotope dilution

The analysis of total dissolved metals for this expedition has been described previously (Kell et al. 2024). Briefly, seawater collected shipboard by pressure-filtering X-Niskin bottles through an acid-washed 142 mm, 0.2 µm polyethersulfone Supor membrane filter (Pall) within 3 hours of rosette recovery using high purity (99.999%) N<sub>2</sub> gas and stored at 4°C. All sample collection occurred shipboard within an on-deck trace metal clean van. Samples were acidified to pH 1.7 with high purity HCl (Optima) within 7 months of collection and were stored acidified at room temperature for over 1 year prior to analysis. This extended acidification time was used to counteract any loss of metal due to adsorption to the bottle walls (Jensen et al. 2020b),

Quantification of total dissolved Fe, Mn, Ni, Cu, Zn, and Cd was performed using isotope dilution. Acidified seawater samples were spiked with a stable isotope spike solution artificially enriched in <sup>57</sup>Fe, <sup>61</sup>Ni, <sup>65</sup>Cu, <sup>67</sup>Zn, and <sup>110</sup>Cd (Oak Ridge National Laboratory). Concentrations and spike ratios were verified by ICP-MS using a multi-element standard curve (SPEX

CertiPrep). Preconcentration of spiked seawater samples for total dissolved metal analysis was performed using the automated solid phase extraction system seaFAST-pico (Elemental Scientific) in offline concentration mode with an initial volume of 15mL and elution volume of 500μL (Rapp et al. 2017; Wuttig et al. 2019). Following preconcentration, multielemental quantitative analysis was performed using an iCAP-Q inductively coupled plasma-mass spectrometer (ICP-MS) (Thermo Scientific). Concentrations of Mn, Fe, Ni, Cu, Zn and Cd were determined using a six-point external standard curve of a multi-element standard (SPEX CertiPrep), diluted to range from 1-10 ppb in 5% nitric acid. An indium standard (SPEX CertiPrep) was similarly added to these standard stocks, diluted to range 1-10 ppb. Instrument injection blanks consisted of 5% nitric acid in Milli-Q. Standard curve R² values were ≥0.98 for all metals monitored. Method accuracy and precision were assessed using the 2009 GEOTRACES coastal surface seawater (GSC) standard (n = 8; Supplementary Table 7), which produced values consistent with consensus results (Kell et al. 2024).

### 5.8 Macronutrient, pigment, and Fv/Fm analyses

Seawater for macronutrient (silicate, phosphate, nitrate, and nitrite) analyses were filtered through 0.2 µm pore-size Supor membrane filters and frozen at sea in acid-washed 60-mL high-density polyethylene bottles until analysis. Macronutrient analyses were conducted by nutrient autoanalyzer (Technicon Autoanalyzer II) by Joe Jennings at Oregon State University. The chemotaxonomic distribution of phytoplankton pigments was determined using HPLC as described previously (DiTullio et al. 2003). Photosynthetic efficiency of photosystem II (Fv/Fm) was measured using a Phyto PAM phytoplankton analyzer (Walz, Effeltrich, Germany) as described previously (Schanke et al. 2021).

### 5.9 Bacterial abundance

One ml samples for heterotrophic prokaryotes abundance (HPA) analysis were fixed for 10 min with a mix of paraformaldehyde and glutaraldehyde (1% and 0.05% final concentration, respectively), frozen in liquid nitrogen and stored at  $-80^{\circ}$ C until analysis. After thawing, samples were stained with SYBR Green (Invitrogen Milan, Italy) using  $10^{-3}$  dilution of stock solution for 15 min at room temperature. Cell concentrations were assessed using a FACSVerse flow cytometer (BD BioSciences Inc., Franklin Lakes, USA) equipped with a 488 nm Ar laser and standard set of optical filters. FCS Express software was used for analyzing the data and HP were discriminated from other particles on the basis of scatter and green fluorescence from SYBR Green (Balestra et al. 2011).

# 5.10 ICP-MS analysis and Zn uptake rates using <sup>67</sup>Zn

<sup>67</sup>Zn stable isotope uptake experiments were performed to quantify the movement of dissolved Zn to the particulate phase in units of pmol L<sup>-1</sup> d<sup>-1</sup> (Kell et al. 2024). Briefly, unfiltered seawater was collected using the trace metal rosette over a depth range of 10 – 600 m into 250mL trace metal clean polycarbonate bottles. Bottles were spiked with <sup>67</sup>Zn such that the total added (spiked) concentration of Zn was 2 nM. Immediately after spiking, incubation bottles were sealed, inverted to mix, and transferred to flow-through on-deck incubators for 24hr. Biomass was collected after 24hr by vacuum filtering each incubation sample at 34.5 kPa (5 psi) onto an acid-cleaned 3μm pore-size acrylic copolymer Versapore filter (Pall) mounted on an acid-cleaned plastic filtration rig. Sample filters were retrieved from storage at -80°C, removed from cryovials using plastic acid-washed forceps, and transferred into trace metal clean 15 mL PFA vials with 4 mL of 5% HNO<sub>3</sub> (Optima) containing a 1 ppb Indium (In) internal standard. Filters

were digested for ~3.5h at 140°C using a HotBlock® heating block (Environmental Express, USA) before they were removed and discarded. After evaporating the remaining solution to just dryness, the residue was resuspended in 2 mL of 5% HNO<sub>3</sub> (Optima) by light vortexing. Process blank filters were digested and processed as sample filters were. This experiment was also carried out using <sup>110</sup>Cd as a tracer of Cd uptake in separate incubation bottles (data not shown here). Digests were analyzed in duplicate by ICP-MS using a Thermo ICAP-Q plasma mass spectrometer calibrated to a multi-element standard curve (Spex Certiprep) over a range of 1 – 20 ppb. Natural Cd and Zn isotope abundances of the standards were assumed to calculate concentrations of <sup>110</sup>Cd, <sup>111</sup>Cd, <sup>114</sup>Cd, <sup>67</sup>Zn, <sup>66</sup>Zn, and <sup>68</sup>Zn. Total Zn uptake (pmol L<sup>-1</sup> d<sup>-1</sup>) was calculated using particulate <sup>67</sup>Zn and total water column dZn measurements as described previously (Cox et al. 2014). The particulate metal measurements captured contributions from the active transport of metal into cells, nonspecific metal adsorption to cell surfaces, metal adsorption to non-living particulate organic matter, and metal adsorption to particulate inorganic matter, though we expect active transport into cells to dominate the measured particulate isotopic signal due to the high abundance of actively growing autotrophic cells in the photic zone observed in Southern Ocean during austral summer. Particulate Zn:P measurements were calculated using particulate Zn measured on Cd-spiked filters and thus do not include any pZn contribution from Zn tracer addition. Particulate phosphorus concentrations were measured by ICP-MS simultaneously and were calibrated to a standard curve ranging from 100 to 3200 ppb using a 1 ppm certified P stock (Alfa Aesar Specpure). All SPEX and P standard curves had R<sup>2</sup> values > 0.99. The Zn stock solution used in the incubation experiments was similarly analyzed by ICP-MS to confirm that the stock was not Fe contaminated—this analysis showed that less than 2.3 pM (which was near the instrument blank level for this analysis) of iron was added for

every 2 nM of zinc, far less than needed to stimulate phytoplankton to the extent observed in our experiments.

592

593

# 5.11 Shipboard incubation experiments

Incubation experiments were conducted at station 27 (-74.9870°N, 165.8898°E). Raw surface seawater was pumped directly into a cleanroom container van, collected into acidcleaned 50L carboys, and dispensed into acid-washed 1L polycarbonate bottles using a trace metal sampling system with acid-washed polypropylene tubing and a teflon diaphragm pump. Incubation bottles were first rinsed with seawater then filled. Seawater was collected at 16:05 UTC. Triplicate incubation bottles were amended with +Fe (1 nM), +Zn (2 nM) and +Fe+Zn, sealed, and placed into a flow-through on-deck incubator with light screens that shaded the incubator to 20% percent ambient surface irradiance. Incubations were sampled at 0, 48, 96, and 144 hours (corresponding to T0, T2, T4, and T6 timepoints) for analysis by filtering onto GFF filters for chlorophyll (all time points, biological triplicates), pigment analyses (T6, biological triplicates), and proteomic analyses (T6, pooled biological triplicates). Chlorophyll was extracted immediately, otherwise samples were frozen at -80°C until further analyses, with pigment and protein samples kept in -80°C freezers, liquid nitrogen dewars, or dry ice coolers at all times during transport back to the laboratories. All amendments and sampling were conducted in a positive-pressure, clean room van with laminar flow hoods and plastic sheeting to minimize trace-metal contamination.

612

613

### 5.12 Metaproteomic analysis

Water column metaproteomic biomass was collected onto 0.2, 3, and 51 µm pore-size filters ("field filters") using in-situ battery operated McLane pumps. Half of each field filter was processed for metaproteomic analysis. Incubation metaproteomic biomass was serially filtered through a 5µm pore prefilter followed by a 142mm GFF filter. Three-fourths of each GFF filter was used for subsequent metaproteomic analysis of the incubations. All filters were frozen at -80°C and stored until laboratory extraction. To extract proteins, filters were placed into extraction buffer (1% SDS, 0.1M Tris/HCL pH 7.5, 10mM EDTA). 8 mL of buffer was used for each field filter, and 15 mL of buffer was used for each GFF incubation filter. All reagents were made with HPLC-grade water. Samples were heated at 95°C for 10 minutes and shaken at room temperature for 30 minutes. Filters were removed and protein extracts were filtered through 5.0 μm Millex low protein binding filters (Merck Millipore #SLSV025LS). Millex filters were rinsed with 1 mL of extraction buffer to ensure no loss of protein. Samples were then spun for 30 minutes at 3220 rcf in an Eppendorf 5810 centrifuge. The supernatant was transferred to Vivaspin 5K MWCO ultrafiltration columns (Sartorius #VS0611). Protein extracts were concentrated to approximately 300 µL, washed with 1 mL of extraction buffer, and transferred to a 2 mL ethanol-washed microtube (all tubes from this point on are ethanol-washed). Vivaspin columns were rinsed with small volumes of protein extraction buffer to remove all concentrated protein and samples were brought up to 400 µL with extraction buffer. Samples were incubated with 2 μL benzonase nuclease (EMD Millipore 70746-3) for 30 minutes at 37°C. Extracted proteins were purified from SDS detergent, reduced, alkylated and digested

with trypsin while embedded within a polyacrylamide tube gel, using a modified, previously published method (Lu and Zhu 2005). A gel premix was made by combining 1 M Tris HCL (pH 7.5) and 40% Bis-acrylamide L 29:1 (Acros Organics) at a ratio of 1:3. The premix (103 μL) was

combined with 50-100 µg of the extracted protein sample, Tris-EDTA, 7 µL 1% APS and 3 µL of TEMED (Acros Organics) to a final volume of 200 µL. After 1 hour of polymerization at room temperature, 200 μL of gel fix solution (50% ETOH, 10% acetic acid in LC/MS grade water) was added to the top of the gel and incubated at room temperature for 20 minutes. Liquid was then removed and the tube gel was transferred into a new 1.5 mL microtube containing 1.2 mL of gel fix solution before incubating at room temperature, 350 rpm in a Thermomixer R (Eppendorf) for 1 h. Gel fix solution was removed and replaced with 1.2 mL of destain solution (50% MeOH, 10% acetic acid in LC/MS grade water) and incubated at 350 rpm, room temperature for 2 h. Liquid was removed, gels were cut up into 1 mm cubes and added back to tubes containing 1 mL of 50:50 acetonitrile:25 mM ammonium bicarbonate (ambic) and incubated for 1 h, 350 rpm at room temperature. Liquid was removed and replaced with fresh 50:50 acetonitrile:ambic solution and incubated at 16°C, 350 rpm overnight. The above step was repeated for 1 hour the following morning. Gel pieces were then dehydrated twice in 800 µl of acetonitrile for 10 min at room temperature and dried for 10 min in a ThermoSavant DNA110 speedvac after removing the solvent. Proteins were reduced in 600 µL of 10 mM DTT, 25 mM ambic at 56°C, 350 rpm for 1 h. The volume of unabsorbed DTT solution was measured prior to removal. Gel pieces were washed with 25 mM ambic, and 600 µL of 55 mM iodoacetamide was added to alkylate proteins at RT, 350 rpm for 1 h. Gel cubes were then washed with 1 mL ambic for 20 minutes, 350 rpm at RT. Acetonitrile (1mL) dehydrations and speedvac drying were repeated as described above. Trypsin (Promega #V5280) was added in an appropriate volume of 25 mM ambic to rehydrate and submerse gel pieces at a concentration of 1:20 µg trypsin:protein. Proteins were digested overnight at 350 rpm, 37°C. Unabsorbed solution was removed and transferred to a new tube. 50 µl of peptide extraction buffer (50% acetonitrile, 5% formic acid in

water) was added to gels, incubated for 20 min at RT, then centrifuged at 14,100 x g for 2 min. The supernatant was collected and combined with the corresponding unabsorbed solution. The above peptide extraction step was repeated again, combining corresponding supernatants. Combined digested peptides were centrifuged at 14,100 x g for 20 minutes, supernatants transferred into a new tube and dehydrated down to approximately 20 μL in the speedvac. Total digested peptides were quantified (Bio-Rad DC protein assay, Hercules, CA) with BSA as a standard. Peptides were then diluted in 2% acetonitrile, 0.1% formic acid in LC/MS grade water to a concentration of 1μg/μL for storage until analysis. All water used in the tube gel digestion protocol was LC/MS grade, and all plastic microtubes were ethanol rinsed and dried prior to use.

Purified peptides were diluted to  $0.1~\mu g~\mu l^{-1}$  and  $20~\mu l~(2~\mu g)$  was injected onto a Dionex UltiMate 3000 RSLCnano LC system (Thermo Fisher Scientific) with an additional RSLCnano pump run in online two-dimensional active modulation mode coupled to a Thermo Fusion Orbitrap mass spectrometer as described previously (McIlvin and Saito 2021).

A translated metatranscriptome (see below) was used as a reference protein database and peptide spectra matches were performed using the SEQUEST algorithm within Proteome Discoverer v.2.1 (Thermo Fisher Scientific) with a fragment tolerance of 0.6 Da and parent tolerance of 10 ppm. Identification criteria consisted of a peptide threshold of 95% (false discovery rate (FDR) = 0.1%) and protein threshold of 99% (1 peptide minimum, FDR = 0.8%) in Scaffold v.5 (Proteome Software) resulting in 5,387 proteins identified in the incubation experiment and 27,924 proteins identified in the water column. To avoid double-counting mass spectra, exclusive spectral counts were used for the downstream proteomic analysis. Exclusive spectral counts were normalized using the normalized spectral abundance factor (NSAF) calculation (Zhang et al. 2010) to allow for a comparison of protein abundance across samples

while remaining consistent with the metatranscriptomic procedure, see Cohen et al. 2021 for details. Counts associated with redundant ORFs (sharing identical taxonomic and functional assignments) were summed together. The stand-alone command line application BLAST+ from the National Center for Biotechnology Information (NCBI) was used to identify proteins of interest in the metaproteomic data. Metaproteomes were BLAST searched (E = 5e-5) against the known sequences of proteins of interest acquired from annotated proteomic databases (Supplementary Table 4) and combined with further annotation data based on contig ID (see below).

## 5.13 Metatranscriptomic analysis

RNA sequencing was performed using the Illumina HiSeq platform. Transcriptomic assemblies were generated for biomass collected using McLane pumps filtered through 0.2, 3, and 51 µm pore-size filters. In order to enrich metatranscriptomic libraries derived from 0.2 µm filters in prokaryotic transcripts and libraries derived from 3 µm and 51 µm filters in eukaryotic transcripts, 0.2 µm libraries were generated from total rRNA-depleted mRNA and 3 µm and 51 µm libraries were generated from polyA mRNA. Total RNA was extracted from 0.2 µm, 3 µm, and 51µm filters using Macherey-Nagel a NucleoMag RNA kit (Macherey-Nagel GmbH & Co.KG). Cleared lysate was loaded into a 96 deep-well plate and put on an epMotion 5075 TMX liquid handler to complete the RNA extraction following the Machery-Nagel standard protocol. For 3 µm and 51 µm samples with total RNA greater than 1 µg, 800 ng of total RNA was used for preparing poly A libraries with an Illumina Stranded mRNA Prep Ligation kit (Illumina), following the manufacturer's protocol. For the 3 µm and 51 µm samples with total RNA less than 1 µg, 20 ng of total RNA was used as input for the SMART-Seq v4 Ultra Low Input RNA

kit (Takara Bio USA. Inc), which converts poly(A) RNA to full-length cDNA using a modified oligo (dT) primer with simultaneous cDNA amplification. The resulting double-stranded cDNA was then fragmented using a Covaris E210 system with the target size of 300bp. Libraries were prepared from fragmented double-stranded cDNA using an Illumina Stranded mRNA Prep Ligation kit (Illumina). For RNA obtained from 0.2 μm filters, ribosomal RNA was removed using a riboPOOL Seawater Kit (Galen Laboratory Supplies, North Haven, Connecticut, USA). The riboPOOL Seawater Kit is a customized mixture of Removal Solutions: Pan-Prokaryote riboPOOL, Pan-Plant riboPOOL and Pan-Mammal in a ratio of 6:1:1. The rRNA-depleted total RNA was used for cDNA synthesis by Ovation RNA-Seq System V2 (TECAN, Redwood City, USA). Double stranded cDNA was then prepared for the libraries using an Illumina Stranded mRNA Prep Ligation kit (Illumina). Ampure XP beads (Beckman Coulter) were used for final library purification. Library quality was analyzed on a 2200 TapeStation System with an Agilent High Sensitivity DNA 1000 ScreenTape System (Agilent Technologies, Santa Clara, CA, USA). Resulting libraries were subjected to paired-end Illumina sequencing via NovaSeq S4. The input paired-end fastq sequences are trimmed of sequencing adapters, primers and low quality bases by using either BLASTN (NCBI, v2.2.25) (Altschul et al. 1990) or trimmomatic, v0.36 (Bolger et al. 2014). The trimmed paired and unpaired sequences were then depleted of rRNA sequences with riboPicker v0.4.3 (Schmieder et al. 2012). The command-line program clc assembler, v5.2.1 (Qiagen) was used to assemble processed sequences into contigs and ORFs were identified by FragGeneScan, v1.31 (Rho et al. 2010). The trimmed sequences were mapped to the predicted ORFs using the command-line program clc mapper, v5.2.1 (Qiagen) to generate mapped raw read counts for each ORF. The raw counts were normalized

initially to RPKM values, to account for variations in inter-sample sequencing depth and the

ORF sequence length (Mortazavi et al. 2008). The RPKM values were subsequently converted to TPM (transcripts per million) units for estimation of the relative RNA abundance among samples (Li and Dewey 2011). The ORFs were annotated for putative function by several programs in parallel using BLASTP against PhyloDB, hidden Markov models PFAM and TIGRFAM by HMMER, v3.3.2 (Eddy 2011), KEGG Ortholog HMM by kofamscan, v1.3.0 (Aramaki et al. 2020), and transmembrane HMM by TMHMM (Krogh et al. 2001). Additional annotations were generated by similarity searches using BLASTP to transporter (PhyloDB), organelle (PhyloDB) and KOG (Tatusov et al. 2003) databases. The ORFs are assigned to the best taxonomic species/group as determined by LPI (Lineage Probability Index) analysis (Podell and Gaasterland 2007). The final list of curated ORFs was generated by removing ORFs with low mapping coverage (< 50 reads total over all samples) and with no BLAST hits and no known domains.

### 5.14 Statistical analysis and data visualization

ANOVA and Dunnett tests were performed using MATLAB 2019a. Statistics are summarized in **Supplementary Table 3**. Figures were made using matplotlib (version 3.5.0), Ocean Data View (version 5.5.2), Excel (2019), and RStudio (version 1.3.1093). Color palettes used in Ocean Data View section plots (<a href="https://doi.org/10.5281/zenodo.1243862">https://doi.org/10.5281/zenodo.1243862</a>) are inverse "roma" for trace metal concentrations, "thermal" for Zn and Cd uptake rates, and "algae" for chlorophyll fluorescence (Crameri 2023).

### Data Availability

| CICLOPS (NBP18-01) conductivity-temperature-depth (CTD) hydrography data                                                                                     |
|--------------------------------------------------------------------------------------------------------------------------------------------------------------|
| including pressure, temperature, total dissolved oxygen, conductivity, fluorescence, and beam                                                                |
| transmission ( <a href="https://doi.org/10.1575/1912/bco-dmo.783911.1">https://doi.org/10.1575/1912/bco-dmo.783911.1</a> ) and total dissolved metal, Zn and |
| Cd uptake rate, macronutrient, and pigment datasets are available through the NSF Biological                                                                 |
| and Chemical Oceanography Data Management Office (BCO-DMO) repository                                                                                        |
| ( <u>https://doi.org/10.7284/907753</u> ). Underway pCO <sub>2</sub> data collected during cruise NBP1801 are                                                |
| available through R2R at <a href="https://doi.org/10.7284/139318">https://doi.org/10.7284/139318</a> . The mass spectrometry global                          |
| proteomics data for CICLOPS bottle incubations and water column analyses have been deposited                                                                 |
| with the ProteomeXchange Consortium through the PRIDE repository under the project name                                                                      |
| "Zinc-iron co-limitation of natural marine phytoplankton assemblages in coastal Antarctica" with                                                             |
| project accession number PXD037056                                                                                                                           |
| (https://www.ebi.ac.uk/pride/archive/projects/PXD037056). This data is accessible for review by                                                              |
| using the following login information: username reviewer_pxd037056@ebi.ac.uk, password:                                                                      |
| lFdOUoEb. The translated transcriptome used for spectrum to peptide matching has been                                                                        |
| deposited in the National Center for Biotechnology Information sequence read archive under                                                                   |
| BioProject accession no. PRJNA890306                                                                                                                         |
| (https://www.ncbi.nlm.nih.gov/bioproject/?term=PRJNA890306) and RNA-Seq BioSample                                                                            |
| accession nos. SAMN31286421-SAMN31286522                                                                                                                     |
| (https://www.ncbi.nlm.nih.gov/biosample/?term=SAMN31286421).                                                                                                 |
|                                                                                                                                                              |

# Acknowledgements

We thank the captain, crew, marine technicians and science party of RVIB Nathaniel B.

Palmer for their support and contributions to the success of the NBP18-01 cruise. We thank

774 Natalie Cohen for assistance and training with the SeaFAST. We thank Veronique Oldham for 775 assistance with trace metal sampling. This work was funded by the National Science Foundation 776 (2125063, 1643684, 1924554) and the Simons Foundation to M.A.S.; NSF-PLR 1643845 to 777 R.B.D, NSF-CO (2123055) to M.A.S. and A.V.S., and the National Institutes of Health 778 (GM135709-01A1 to M.A.S). 779 780 **Author contributions statement** 781 All authors contributed to data acquisition and analysis. RMK, NLS, LEL, RJC, DR, DMM, 782 MRM, FB, RBD, GRD and MAS implemented the shipboard incubation study. RMK, MMB, 783 RJC, DR, TJH, and AVS contributed to the formal analysis. NLS, LEL, FB, OM, RF, and GRD 784 contributed to pigment datasets and interpretations. CB provided bacterial abundance data. RBD 785 contributed DIC and in situ pCO<sub>2</sub> data. AVS contributed analyses and discussion regarding 786 historical pCO<sub>2</sub> data and Zn-C growth limitation estimates. AEA contributed the 787 metatranscriptome reference database used for proteomic analyses. RMK and MAS wrote the

original draft. RMK, MAS, GRD, AVS, TJH, and RBD contributed to review and editing. All

790

791

792

788

789

## **Competing interests statement**

The authors declare no competing interests.

authors approved the final submitted manuscript.

793

794

795

# 797 Figures

**Figure 1**. Temporal biogeochemistry of Terra Nova Bay and characterization of the experimental site at Station 27. (a) Sampling locations over the Ross Sea shelf in Terra Nova Bay, Antarctica. (b) Location of station 27 (red star) and surrounding seawater pCO<sub>2</sub> measured over a three-day transit northwards represented in color scale. (c) pCO<sub>2</sub> measured over time

within TNB during the three-day transit shown in (b). The vertical red line denotes the pCO<sub>2</sub> level at the time of initial seawater collection at station 27. (d) Total chlorophyll fluorescence, (e) fucoxanthin (fuco), (f) 19' hexanoyloxyfucoxanthin (19'-hex), (g) total dissolved Zn, (h) total Zn uptake rates, and (i) total dissolved Fe measured in the upper 250 m represented on a color scale. Station data is presented in order of sampling date, from the earliest (Stn 22, early January) to the latest (Stn 79, late February). The data gap between January 13-23 occurred when the ship was unable to sample due to icebreaking duties for the McMurdo Station resupply ship. Stations indicated in (a) are those where the trace metal rosette (TMR) was deployed; pigment data was supplemented with additional TNB stations using a CTD (Table S2). Depth profiles of (j) chlorophyll a, (k) the pigments fuco and 19'hex, (l) total dissolved inorganic carbon (DICT), (m) total dissolved Zn, (n) total Zn uptake rates, and (o) the macronutrients nitrate+nitrite (N+N), phosphate (P), and silicate (Si) at the study site. Panels (d),(g),(h) and (i) were originally presented in Kell et al. (2024) and are reprised here to introduce the environmental context of the study site.

Figure 2. Evidence for Zn co-limitation with Fe in bottle incubations. (a) Chlorophyll *a* and (b) total DIC (DICT) at T0 (day 0) and in each treatment at T6 (day 6). Significant differences among groups were found using one-way ANOVA and post-hoc Dunnett test (\*\*\* p 

Figure 3. Metatranscriptomic and metaproteomic detection of Zn- and Fe-related proteins of interest at the experimental site. Depth profiles of summed NSAF-normalized protein spectral counts of (a) RUBISCO, (b) ZCRP-A, (c) iron starvation induced proteins (ISIPs), (d) ZCRP-B, (e) ZIP, (f) Theta CA, and (g) Delta CA detected from proteomic analysis of each filter size fraction (0.2, 3 and 51μm). (h) TPM-normalized transcript read counts of CDCA. (i) Stacked pie charts depicting relative community composition for proteins of interest for euphotic (

Figure 4. The partial pressure of CO<sub>2</sub> (pCO<sub>2</sub>) and associated phytoplankton responses from this study and the literature. (a) Pre-industrial and decadal surface ocean pCO<sub>2</sub> reconstructions plotted as violin plots, with a running black line through the median values. The atmospheric curve is a composite of ice core data (dashed yellow line (Etheridge et al. 1996)) and the Mauna Loa record (solid yellow line (Keeling et al. 1976)). An estimated thresholds for zinc-limited growth is plotted as the median of previous laboratory results (259 μatm, dark green line; see Methods), and is compared to the *in situ* results of this study (220 μatm, light green line). (b) Data in (a) plotted as a histogram comparing preindustrial and modern (2010) pCO<sub>2</sub> values, with the same pCO<sub>2</sub> levels indicated. (c) Global map of surface ocean pCO<sub>2</sub> plotted using GLODAPv2.2022 data (Lauvset et al. 2022). (b) Percentage of the ocean surface less than 250 μatm pCO<sub>2</sub> as a function of time. Surface ocean pCO<sub>2</sub> reconstructions taken from Jiang et al. 2023.

| 868 | References                                                                                         |
|-----|----------------------------------------------------------------------------------------------------|
| 869 | Allen, A. E. and others. 2008. Whole-cell response of the pennate diatom Phaeodactylum             |
| 870 | tricornutum to iron starvation. Proc. Natl. Acad. Sci. 105: 10438–10443.                           |
| 871 | doi:10.1073/pnas.0711370105                                                                        |
| 872 | Altschul, S. F., W. Gish, W. Miller, E. W. Myers, and D. J. Lipman. 1990. Basic local alignment    |
| 873 | search tool. J. Mol. Biol. doi:10.1016/S0022-2836(05)80360-2                                       |
| 874 | Aramaki, T., R. Blanc-Mathieu, H. Endo, K. Ohkubo, M. Kanehisa, S. Goto, and H. Ogata.             |
| 875 | 2020. KofamKOALA: KEGG Ortholog assignment based on profile HMM and adaptive                       |
| 876 | score threshold A. Valencia [ed.]. Bioinformatics 36: 2251–2252.                                   |
| 877 | doi:10.1093/bioinformatics/btz859                                                                  |
| 878 | Arrigo, K. R., K. E. Lowry, and G. L. van Dijken. 2012. Annual changes in sea ice and              |
| 879 | phytoplankton in polynyas of the Amundsen Sea, Antarctica. Deep Sea Res. Part II Top.              |
| 880 | Stud. Oceanogr. <b>71–76</b> : 5–15. doi:10.1016/j.dsr2.2012.03.006                                |
| 881 | Baars, O., and P. L. Croot. 2011. The speciation of dissolved zinc in the Atlantic sector of the   |
| 882 | Southern Ocean. Deep Sea Res. Part II Top. Stud. Oceanogr. 58: 2720–2732.                          |
| 883 | doi:10.1016/j.dsr2.2011.02.003                                                                     |
| 884 | Balestra, C., L. Alonso-Sáez, J. M. Gasol, and R. Casotti. 2011. Group-specific effects on coastal |
| 885 | bacterioplankton of polyunsaturated aldehydes produced by diatoms. Aquat. Microb.                  |
| 886 | Ecol. doi:10.3354/ame01486                                                                         |
| 887 | Behnke, J., and J. LaRoche. 2020. Iron uptake proteins in algae and the role of Iron Starvation-   |
| 888 | Induced Proteins (ISIPs). Eur. J. Phycol. 55: 339–360.                                             |
| 889 | doi:10.1080/09670262.2020.1744039                                                                  |

| 890 | Bender, S. J. and others. 2018. Colony formation in Phaeocystis antarctica: connecting molecular |
|-----|--------------------------------------------------------------------------------------------------|
| 891 | mechanisms with iron biogeochemistry. Biogeosciences 15: 4923-4942. doi:10.5194/bg-              |
| 892 | 15-4923-2018                                                                                     |
| 893 | Blaby-Haas, C. E., and S. S. Merchant. 2012. The ins and outs of algal metal transport. Biochim. |
| 894 | Biophys. Acta - Mol. Cell Res. <b>1823</b> : 1531–1552. doi:10.1016/j.bbamcr.2012.04.010         |
| 895 | Bolger, A. M., M. Lohse, and B. Usadel. 2014. Trimmomatic: a flexible trimmer for Illumina       |
| 896 | sequence data. Bioinformatics 30: 2114–2120. doi:10.1093/bioinformatics/btu170                   |
| 897 | Browning, T. J., and C. M. Moore. 2023. Global analysis of ocean phytoplankton nutrient          |
| 898 | limitation reveals high prevalence of co-limitation. Nat. Commun. 14: 5014.                      |
| 899 | doi:10.1038/s41467-023-40774-0                                                                   |
| 900 | Bruland, K. W. 1980. Oceanographic distributions of cadmium, zinc, nickel, and copper in the     |
| 901 | North Pacific. Earth Planet. Sci. Lett. 47: 176-198. doi:10.1016/0012-821X(80)90035-7            |
| 902 | Bruland, K. W. 1989. Complexation of zinc by natural organic ligands in the central North        |
| 903 | Pacific. Limnol. Oceanogr. 34: 269–285. doi:10.4319/lo.1989.34.2.0269                            |
| 904 | Buitenhuis, E. T., K. R. Timmermans, and H. J. W. de Baar. 2003. Zinc-bicarbonate colimitation   |
| 905 | of Emiliania huxleyi. Limnol. Oceanogr. 48: 1575–1582. doi:10.4319/lo.2003.48.4.1575             |
| 906 | Chavez, F. P., J. Sevadjian, C. Wahl, J. Friederich, and G. E. Friederich. 2018. Measurements of |
| 907 | pCO2 and pH from an autonomous surface vehicle in a coastal upwelling system. Deep               |
| 908 | Sea Res. Part II Top. Stud. Oceanogr. 151: 137–146. doi:10.1016/j.dsr2.2017.01.001               |
| 909 | Coale, K. H., R. Michael Gordon, and X. Wang. 2005. The distribution and behavior of             |
| 910 | dissolved and particulate iron and zinc in the Ross Sea and Antarctic circumpolar current        |
| 911 | along 170°W. Deep Sea Res. Part Oceanogr. Res. Pap. <b>52</b> : 295–318.                         |
| 912 | doi:10.1016/j.dsr.2004.09.008                                                                    |

| 913 | Coale, K. H., X. Wang, S. J. Tanner, and K. S. Johnson. 2003. Phytoplankton growth and            |
|-----|---------------------------------------------------------------------------------------------------|
| 914 | biological response to iron and zinc addition in the Ross Sea and Antarctic Circumpolar           |
| 915 | Current along 170°W. Deep Sea Res. Part II Top. Stud. Oceanogr. <b>50</b> : 635–653.              |
| 916 | doi:10.1016/S0967-0645(02)00588-X                                                                 |
| 917 | Cochlan, W. P., D. A. Bronk, and K. H. Coale. 2002. Trace metals and nitrogenous nutrition of     |
| 918 | Antarctic phytoplankton: experimental observations in the Ross Sea. Deep Sea Res. Part            |
| 919 | II Top. Stud. Oceanogr. <b>49</b> : 3365–3390. doi:10.1016/S0967-0645(02)00088-7                  |
| 920 | Cox, A. D., A. E. Noble, and M. A. Saito. 2014. Cadmium enriched stable isotope uptake and        |
| 921 | addition experiments with natural phytoplankton assemblages in the Costa Rica                     |
| 922 | Upwelling Dome. Mar. Chem. 166: 70-81. doi:10.1016/j.marchem.2014.09.009                          |
| 923 | Crameri, F. 2023. Scientific colour maps.doi:10.5281/ZENODO.1243862                               |
| 924 | Crawford, D. W. and others. 2003a. Influence of zinc and iron enrichments on phytoplankton        |
| 925 | growth in the northeastern subarctic Pacific Limitation of biomass and primary                    |
| 926 | production of phyto-plankton by dissolved iron (Fe) in high-nutrient, low-chlo.                   |
| 927 | Crawford, D. W. and others. 2003b. Influence of zinc and iron enrichments on phytoplankton        |
| 928 | growth in the northeastern subarctic Pacific. Limnol. Oceanogr. 48: 1583–1600.                    |
| 929 | doi:10.4319/lo.2003.48.4.1583                                                                     |
| 930 | Cullen, J. T., T. W. Lane, F. M. M. Morel, and R. M. Sheerell. 1999. Modulation of cadmium        |
| 931 | uptake in phytoplankton by seawater CO2 concentration. Nature. doi:10.1038/46007                  |
| 932 | Cullen, J. T., and R. M. Sherrell. 2005. Effects of dissolved carbon dioxide, zinc, and manganese |
| 933 | on the cadmium to phosphorus ratio in natural phytoplankton assemblages. Limnol.                  |
| 934 | Oceanogr. <b>50</b> : 1193–1204. doi:10.4319/lo.2005.50.4.1193                                    |

| 935 | Cutter, G. A., and K. W. Bruland. 2012. Rapid and noncontaminating sampling system for trace      |
|-----|---------------------------------------------------------------------------------------------------|
| 936 | elements in global ocean surveys. Limnol. Oceanogr. Methods.                                      |
| 937 | doi:10.4319/lom.2012.10.425                                                                       |
| 938 | Dai, M. and others. 2022. Carbon Fluxes in the Coastal Ocean: Synthesis, Boundary Processes,      |
| 939 | and Future Trends. Annu. Rev. Earth Planet. Sci. 50: 593-626. doi:10.1146/annurev-                |
| 940 | earth-032320-090746                                                                               |
| 941 | DeJong, H. B., R. B. Dunbar, D. Mucciarone, and D. A. Koweek. 2015. Carbonate saturation          |
| 942 | state of surface waters in the Ross Sea and Southern Ocean: controls and implications for         |
| 943 | the onset of aragonite undersaturation. Biogeosciences 12: 6881-6896. doi:10.5194/bg-             |
| 944 | 12-6881-2015                                                                                      |
| 945 | Dickson, A. G., C. L. Sabine, and J. R. Christian, 2007. Guide to best practices for ocean CO2    |
| 946 | measurement. North Pacific Marine Science Organization.                                           |
| 947 | DiTullio, G. R. and others. 2000. Rapid and early export of Phaeocystis antarctica blooms in the  |
| 948 | Ross Sea, Antarctica. Nature. doi:10.1038/35007061                                                |
| 949 | DiTullio, G. R., N. Garcia, S. F. Riseman, and P. N. Sedwick. 2007. Effects of iron concentration |
| 950 | on pigment composition in Phaeocystis antarctica grown at low irradiance.                         |
| 951 | Biogeochemistry 83: 71-81. doi:10.1007/s10533-007-9080-8                                          |
| 952 | DiTullio, G. R., M. E. Geesey, A. Leventer, and M. P. Lizotte. 2003. Algal pigment ratios in the  |
| 953 | Ross Sea: Implications for Chemtax analysis of Southern Ocean data, p. 35–51. <i>In</i> .         |
| 954 | DiTullio, G. R., and W. O. Smith. 1996. Spatial patterns in phytoplankton biomass and pigment     |
| 955 | distributions in the Ross Sea. J. Geophys. Res. Oceans 101: 18467–18477.                          |
| 956 | doi:10.1029/96JC00034                                                                             |

| 957 | Dreux Chappell, P., J. Vedmati, K. E. Selph, H. A. Cyr, B. D. Jenkins, M. R. Landry, and J. W. |
|-----|------------------------------------------------------------------------------------------------|
| 958 | Moffett. 2016. Preferential depletion of zinc within Costa Rica upwelling dome creates         |
| 959 | conditions for zinc co-limitation of primary production. J. Plankton Res. 38: 244–255.         |
| 960 | doi:10.1093/plankt/fbw018                                                                      |
| 961 | Eddy, S. R. 2011. Accelerated Profile HMM Searches W.R. Pearson [ed.]. PLoS Comput. Biol.      |
| 962 | 7: e1002195. doi:10.1371/journal.pcbi.1002195                                                  |
| 963 | Edmonds, K. A., M. R. Jordan, and D. P. Giedroc. 2021. COG0523 proteins: a functionally        |
| 964 | diverse family of transition metal-regulated G3E P-loop GTP hydrolases from bacteria to        |
| 965 | man. Metallomics 13. doi:10.1093/mtomcs/mfab046                                                |
| 966 | Ellwood, M. J. 2004. Zinc and cadmium speciation in subantarctic waters east of New Zealand.   |
| 967 | Mar. Chem. doi:10.1016/j.marchem.2004.01.005                                                   |
| 968 | Emms, D. M., and S. Kelly. 2022. SHOOT: phylogenetic gene search and ortholog inference.       |
| 969 | Genome Biol. 23. doi:10.1186/s13059-022-02652-8                                                |
| 970 | Etheridge, D. M., L. P. Steele, R. L. Langenfelds, R. J. Francey, JM. Barnola, and V. I.       |
| 971 | Morgan. 1996. Natural and anthropogenic changes in atmospheric CO 2 over the last              |
| 972 | 1000 years from air in Antarctic ice and firn. J. Geophys. Res. Atmospheres 101: 4115-         |
| 973 | 4128. doi:10.1029/95JD03410                                                                    |
| 974 | Fitzwater, S. E., K. S. Johnson, R. M. Gordon, K. H. Coale, and W. O. Smith. 2000. Trace metal |
| 975 | concentrations in the Ross Sea and their relationship with nutrients and phytoplankton         |
| 976 | growth. Deep-Sea Res. Part II Top. Stud. Oceanogr. 47: 3159-3179. doi:10.1016/S0967-           |
| 977 | 0645(00)00063-1                                                                                |

978 Fourquez, M., M. Bressac, S. L. Deppeler, M. Ellwood, I. Obernosterer, T. W. Trull, and P. W. 979 Boyd. 2020. Microbial Competition in the Subpolar Southern Ocean: An Fe-C Co-980 limitation Experiment. Front. Mar. Sci. 6: 776. doi:10.3389/fmars.2019.00776 981 Franck, V., K. Bruland, D. Hutchins, and M. Brzezinski. 2003. Iron and zinc effects on silicic 982 acid and nitrate uptake kinetics in three high-nutrient, low-chlorophyll (HNLC) regions. 983 Mar. Ecol. Prog. Ser. **252**: 15–33. doi:10.3354/meps252015 984 Fukuda, R., Y. Sohrin, N. Saotome, H. Fukuda, T. Nagata, and I. Koike. 2000. East—west 985 gradient in ectoenzyme activities in the subarctic Pacific: Possible regulation by zinc. 986 Limnol. Oceanogr. **45**: 930–939. doi:10.4319/lo.2000.45.4.0930 987 Harrison, C. S., M. C. Long, N. S. Lovenduski, and J. K. Moore. 2018. Mesoscale Effects on 988 Carbon Export: A Global Perspective. Glob. Biogeochem. Cycles **32**: 680–703. 989 doi:10.1002/2017GB005751 990 Jakuba, R. W., M. A. Saito, J. W. Moffett, and Y. Xu. 2012. Dissolved zinc in the subarctic 991 North Pacific and Bering Sea: Its distribution, speciation, and importance to primary 992 producers. Glob. Biogeochem. Cycles. doi:10.1029/2010GB004004 993 Jensen, E. L., S. C. Maberly, and B. Gontero. 2020a. Insights on the Functions and 994 Ecophysiological Relevance of the Diverse Carbonic Anhydrases in Microalgae. Int. J. 995 Mol. Sci. 21: 2922. doi:10.3390/ijms21082922 996 Jensen, L. T., N. J. Wyatt, W. M. Landing, and J. N. Fitzsimmons. 2020b. Assessment of the 997 stability, sorption, and exchangeability of marine dissolved and colloidal metals. Mar. 998 Chem. **220**: 103754. doi:10.1016/j.marchem.2020.103754 999 Jiang, L. and others. 2023. Global Surface Ocean Acidification Indicators From 1750 to 2100. J. 1000 Adv. Model. Earth Syst. 15: e2022MS003563. doi:10.1029/2022MS003563

1001 Keeling, C. D., R. B. Bacastow, A. E. Bainbridge, C. A. Ekdahl, P. R. Guenther, L. S. 1002 Waterman, and J. F. S. Chin. 1976. Atmospheric carbon dioxide variations at Mauna Loa 1003 Observatory, Hawaii. Tellus Dyn. Meteorol. Oceanogr. 28: 538. 1004 doi:10.3402/tellusa.v28i6.11322 1005 Kell, R. M. and others. 2024. High metabolic zinc demand within native Amundsen and Ross sea 1006 phytoplankton communities determined by stable isotope uptake rate measurements. 1007 Biogeosciences 21: 5685–5706. doi:10.5194/bg-21-5685-2024 1008 Kellogg, R. M. and others. 2022a. Adaptive responses of marine diatoms to zinc scarcity and 1009 ecological implications. Nat. Commun. 2022 131 13: 1–13. doi:10.1038/s41467-022-1010 29603-y 1011 Kellogg, R. M., M. R. McIlvin, J. Vedamati, B. S. Twining, J. W. Moffett, A. Marchetti, D. M. 1012 Moran, and M. A. Saito. 2020. Efficient zinc/cobalt inter-replacement in northeast Pacific 1013 diatoms and relationship to high surface dissolved Co: Zn ratios. Limnol. Oceanogr. 65: 1014 2557–2582. doi:10.1002/lno.11471 1015 Kellogg, R. M., D. M. Moran, M. R. McIlvin, A. V. Subhas, A. E. Allen, and M. A. Saito. 1016 2022b. Lack of a Zn/Co substitution ability in the polar diatom *Chaetoceros neogracile* 1017 RS19. Limnol. Oceanogr. 67: 2265–2280. doi:10.1002/lno.12201 1018 Krogh, A., B. Larsson, G. von Heijne, and E. L. L. Sonnhammer. 2001. Predicting 1019 transmembrane protein topology with a hidden markov model: application to complete 1020 genomes11Edited by F. Cohen. J. Mol. Biol. **305**: 567–580. doi:10.1006/jmbi.2000.4315 1021 Krumhardt, K. M., N. S. Lovenduski, M. D. Iglesias-Rodriguez, and J. A. Kleypas. 2017. 1022 Coccolithophore growth and calcification in a changing ocean. Prog. Oceanogr. 159: 1023 276–295. doi:10.1016/j.pocean.2017.10.007

- 1024 Lampe, R. H. and others. 2018. Different iron storage strategies among bloom-forming diatoms. 1025 Proc. Natl. Acad. Sci. 115. doi:10.1073/pnas.1805243115 1026 Lane, T. W., and F. M. Morel. 2000a. A biological function for cadmium in marine diatoms. 1027 Proc. Natl. Acad. Sci. U. S. A. 97: 4627–31. doi:10.1073/pnas.090091397 1028 Lane, T. W., and F. M. M. Morel. 2000b. Regulation of Carbonic Anhydrase Expression by 1029 Zinc, Cobalt, and Carbon Dioxide in the Marine Diatom Thalassiosira weissflogii. Plant 1030 Physiol. **123**: 345–352. doi:10.1104/pp.123.1.345 1031 Lauvset, S. K. and others. 2022. GLODAPv2.2022: the latest version of the global interior ocean 1032 biogeochemical data product. Earth Syst. Sci. Data 14: 5543-5572. doi:10.5194/essd-14-1033 5543-2022 1034 Lhospice, S. and others. 2017. Pseudomonas aeruginosa zinc uptake in chelating environment is 1035 primarily mediated by the metallophore pseudopaline. Sci. Rep. 7: 17132. 1036 doi:10.1038/s41598-017-16765-9 1037 Li, B., and C. N. Dewey. 2011. RSEM: accurate transcript quantification from RNA-Seq data 1038 with or without a reference genome. BMC Bioinformatics 12: 323. doi:10.1186/1471-1039 2105-12-323 1040 Lohan, M. C., P. J. Statham, and D. W. Crawford. 2002. Total dissolved zinc in the upper water 1041 column of the subarctic North East Pacific. Deep Sea Res. Part II Top. Stud. Oceanogr.
- 1043 Lu, X., and H. Zhu. 2005. Tube-Gel Digestion. Mol. Cell. Proteomics **4**: 1948–1958.

**49**: 5793–5808. doi:10.1016/S0967-0645(02)00215-1

doi:10.1074/mcp.m500138-mcp200

- 1045 Mangoni, O., M. Saggiomo, F. Bolinesi, M. Castellano, P. Povero, V. Saggiomo, and G. R.
- DiTullio. 2019. Phaeocystis antarctica unusual summer bloom in stratified antarctic

| 1047 | coastal waters (Terra Nova Bay, Ross Sea). Mar. Environ. Res.                                   |
|------|-------------------------------------------------------------------------------------------------|
| 1048 | doi:10.1016/j.marenvres.2019.05.012                                                             |
| 1049 | Martin, J. H., S. E. Fitzwater, and R. M. Gordon. 1990. Iron deficiency limits phytoplankton    |
| 1050 | growth in Antarctic waters. Glob. Biogeochem. Cycles 4: 5–12.                                   |
| 1051 | doi:10.1029/GB004i001p00005                                                                     |
| 1052 | Mazzotta, M. G., M. R. McIlvin, D. M. Moran, D. T. Wang, K. D. Bidle, C. H. Lamborg, and M.     |
| 1053 | A. Saito. 2021. Characterization of the metalloproteome of Pseudoalteromonas (BB2-              |
| 1054 | AT2): biogeochemical underpinnings for zinc, manganese, cobalt, and nickel cycling in a         |
| 1055 | ubiquitous marine heterotroph. Metallomics 13. doi:10.1093/mtomcs/mfab060                       |
| 1056 | McIlvin, M. R., and M. A. Saito. 2021. Online Nanoflow Two-Dimension Comprehensive              |
| 1057 | Active Modulation Reversed Phase-Reversed Phase Liquid Chromatography High-                     |
| 1058 | Resolution Mass Spectrometry for Metaproteomics of Environmental and Microbiome                 |
| 1059 | Samples. J. Proteome Res. 20: 4589–4597. doi:10.1021/acs.jproteome.1c00588                      |
| 1060 | Middag, R., H. J. W. De Baar, and K. W. Bruland. 2019. The Relationships Between Dissolved      |
| 1061 | Zinc and Major Nutrients Phosphate and Silicate Along the GEOTRACES GA02                        |
| 1062 | Transect in the West Atlantic Ocean. Glob. Biogeochem. Cycles 33: 63-84.                        |
| 1063 | doi:10.1029/2018GB006034                                                                        |
| 1064 | Milner, M. J., J. Seamon, E. Craft, and L. V. Kochian. 2013. Transport properties of members of |
| 1065 | the ZIP family in plants and their role in Zn and Mn homeostasis. J. Exp. Bot. 64: 369–         |
| 1066 | 381. doi:10.1093/jxb/ers315                                                                     |
| 1067 | Moore, C. M. and others. 2013. Processes and patterns of oceanic nutrient limitation. Nat.      |
| 1068 | Geosci. 6: 701–710. doi:10.1038/ngeo1765                                                        |

| 1069 | Morel, F. M. M., J. R. Reinfelder, S. B. Roberts, C. P. Chamberlain, J. G. Lee, and D. Yee. 1994 |
|------|--------------------------------------------------------------------------------------------------|
| 1070 | Zinc and carbon co-limitation of marine phytoplankton. Nature <b>369</b> : 740–742.              |
| 1071 | doi:10.1038/369740A0                                                                             |
| 1072 | Mortazavi, A., B. A. Williams, K. McCue, L. Schaeffer, and B. Wold. 2008. Mapping and            |
| 1073 | quantifying mammalian transcriptomes by RNA-Seq. Nat. Methods 5: 621–628.                        |
| 1074 | doi:10.1038/nmeth.1226                                                                           |
| 1075 | Obernosterer, I., M. Fourquez, and S. Blain. 2015. Fe and C co-limitation of heterotrophic       |
| 1076 | bacteria in the naturally fertilized region off the Kerguelen Islands. Biogeosciences 12:        |
| 1077 | 1983–1992. doi:10.5194/bg-12-1983-2015                                                           |
| 1078 | Podell, S., and T. Gaasterland. 2007. DarkHorse: a method for genome-wide prediction of          |
| 1079 | horizontal gene transfer. Genome Biol. 8: R16. doi:10.1186/gb-2007-8-2-r16                       |
| 1080 | Rapp, I., C. Schlosser, D. Rusiecka, M. Gledhill, and E. P. Achterberg. 2017. Automated          |
| 1081 | preconcentration of Fe, Zn, Cu, Ni, Cd, Pb, Co, and Mn in seawater with analysis using           |
| 1082 | high-resolution sector field inductively-coupled plasma mass spectrometry. Anal. Chim.           |
| 1083 | Acta. doi:10.1016/j.aca.2017.05.008                                                              |
| 1084 | Redfield, A. C. 1958. The Biological Control of Chemical Factors in the Environment. Am. Sci.    |
| 1085 | doi:10.2307/27828530                                                                             |
| 1086 | Rho, M., H. Tang, and Y. Ye. 2010. FragGeneScan: predicting genes in short and error-prone       |
| 1087 | reads. Nucleic Acids Res. 38: e191-e191. doi:10.1093/nar/gkq747                                  |
| 1088 | Riebesell, U., D. A. Wolf-Gladrow, and V. Smetacek. 1993. Carbon dioxide limitation of marine    |
| 1089 | phytoplankton growth rates. Nature 361: 249-251. doi:10.1038/361249a0                            |

| 1090 | Robbins, L. L. and others. 2018. Spatial and Temporal Variability of $pCO_2$ , Carbon Fluxes, and   |
|------|-----------------------------------------------------------------------------------------------------|
| 1091 | Saturation State on the West Florida Shelf. J. Geophys. Res. Oceans 123: 6174–6188.                 |
| 1092 | doi:10.1029/2018JC014195                                                                            |
| 1093 | Roshan, S., T. DeVries, J. Wu, and G. Chen. 2018. The Internal Cycling of Zinc in the Ocean.        |
| 1094 | Glob. Biogeochem. Cycles <b>32</b> : 1833–1849. doi:10.1029/2018GB006045                            |
| 1095 | Saito, M. A., and T. J. Goepfert. 2008. Zinc-cobalt colimitation of <i>Phaeocystis antarctica</i> . |
| 1096 | Limnol. Oceanogr. 53: 266–275. doi:10.4319/lo.2008.53.1.0266                                        |
| 1097 | Saito, M. A., T. J. Goepfert, and J. T. Ritt. 2008. Some thoughts on the concept of colimitation:   |
| 1098 | Three definitions and the importance of bioavailability. Limnol. Oceanogr. <b>53</b> : 276–290.     |
| 1099 | doi:10.4319/lo.2008.53.1.0276                                                                       |
| 1100 | Schanke, N. L., F. Bolinesi, O. Mangoni, C. Katlein, P. Anhaus, M. Hoppmann, P. A. Lee, and         |
| 1101 | G. R. DiTullio. 2021. Biogeochemical and ecological variability during the late summer-             |
| 1102 | early autumn transition at an ice-floe drift station in the Central Arctic Ocean. Limnol.           |
| 1103 | Oceanogr. doi:10.1002/lno.11676                                                                     |
| 1104 | Scharek, R., M. A. Van Leeuwe, and H. J. W. De Baar. 1997. Responses of Southern Ocean              |
| 1105 | phytoplankton to the addition of trace metals. Deep-Sea Res. Part II Top. Stud. Oceanogr            |
| 1106 | doi:10.1016/S0967-0645(96)00074-4                                                                   |
| 1107 | Schmieder, R., Y. W. Lim, and R. Edwards. 2012. Identification and removal of ribosomal RNA         |
| 1108 | sequences from metatranscriptomes. Bioinformatics 28: 433–435.                                      |
| 1109 | doi:10.1093/bioinformatics/btr669                                                                   |
| 1110 | Schneider, B., and J. D. Müller. 2018. Biogeochemical transformations in the Baltic Sea:            |
| 1111 | observations through carbon dioxide glasses, Springer.                                              |

1112 Sharma, D., H. Biswas, S. Silori, D. Bandyopadhyay, A. urRahman Shaik, D. Cardinal, M. 1113 Mandeng-Yogo, and D. Ray. 2020. Impacts of Zn and Cu enrichment under ocean 1114 acidification scenario on a phytoplankton community from tropical upwelling system. 1115 Mar. Environ. Res. 155: 104880. doi:10.1016/j.marenvres.2020.104880 1116 Sharp, J. D., D. Pierrot, M. P. Humphreys, J.-M. Epitalon, J. C. Orr, E. R. Lewis, and D. W. R. 1117 Wallace. 2023. CO2SYSv3 for MATLAB.doi:10.5281/ZENODO.3950562 1118 Shim, J., D. Kim, Y. C. Kang, J. H. Lee, S.-T. Jang, and C.-H. Kim. 2007. Seasonal variations in 1119 pCO2 and its controlling factors in surface seawater of the northern East China Sea. Cont. 1120 Shelf Res. 27: 2623–2636. doi:10.1016/j.csr.2007.07.005 1121 Sieber, M., T. M. Conway, G. F. de Souza, C. S. Hassler, M. J. Ellwood, and D. Vance. 2020. 1122 Cycling of zinc and its isotopes across multiple zones of the Southern Ocean: Insights 1123 from the Antarctic Circumnavigation Expedition. Geochim. Cosmochim. Acta. 1124 doi:10.1016/j.gca.2019.09.039 1125 Smith, W. O., A. R. Shields, J. A. Peloquin, G. Catalano, S. Tozzi, M. S. Dinniman, and V. A. 1126 Asper. 2006. Interannual variations in nutrients, net community production, and 1127 biogeochemical cycles in the Ross Sea. Deep-Sea Res. Part II Top. Stud. Oceanogr. 1128 doi:10.1016/j.dsr2.2006.02.014 1129 Sperfeld, E., D. Raubenheimer, and A. Wacker. 2016. Bridging factorial and gradient concepts 1130 of resource co-limitation: towards a general framework applied to consumers J. Grover 1131 [ed.]. Ecol. Lett. 19: 201–215. doi:10.1111/ele.12554 1132 Subhas, A. V., J. F. Adkins, S. Dong, N. E. Rollins, and W. M. Berelson. 2019. The carbonic 1133 anhydrase activity of sinking and suspended particles in the North Pacific Ocean. Limnol. 1134 Oceanogr. 65: 637–651. doi:10.1002/lno.11332

| 1135 | Sun, Y., P. Debeljak, and I. Obernosterer. 2021. Microbial iron and carbon metabolism as     |
|------|----------------------------------------------------------------------------------------------|
| 1136 | revealed by taxonomy-specific functional diversity in the Southern Ocean. ISME J. 15:        |
| 1137 | 2933–2946. doi:10.1038/s41396-021-00973-3                                                    |
| 1138 | Sunda, W. G., and S. A. Huntsman. 1995. Cobalt and zinc interreplacement in marine           |
| 1139 | phytoplankton: Biological and geochemical implications. Limnol. Oceanogr. 40: 1404–          |
| 1140 | 1417. doi:10.4319/lo.1995.40.8.1404                                                          |
| 1141 | Sunda, W. G., and S. A. Huntsman. 2000. Effect of Zn, Mn, and Fe on Cd accumulation in       |
| 1142 | phytoplankton: Implications for oceanic Cd cycling. Limnol. Oceanogr. 45: 1501–1516.         |
| 1143 | doi:10.4319/lo.2000.45.7.1501                                                                |
| 1144 | Sunda, W. G., and S. A. Huntsman. 2005. Effect of CO 2 supply and demand on zinc uptake and  |
| 1145 | growth limitation in a coastal diatom. Limnol. Oceanogr. <b>50</b> : 1181–1192.              |
| 1146 | doi:10.4319/lo.2005.50.4.1181                                                                |
| 1147 | Takahashi, T., S. C. Sutherland, and A. Kozyr. 2020. Global Ocean Surface Water Partial      |
| 1148 | Pressure of CO2 Database (LDEO Database Version 2019): Measurements Performed                |
| 1149 | During 1957-2019 (NCEI Accession 0160492). NOAA Natl. Cent. Environ. Inf.                    |
| 1150 | Tatusov, R. L. and others. 2003. The COG database: an updated version includes eukaryotes.   |
| 1151 | BMC Bioinformatics 4: 41. doi:10.1186/1471-2105-4-41                                         |
| 1152 | Tortell, P. D., M. C. Long, C. D. Payne, AC. Alderkamp, P. Dutrieux, and K. R. Arrigo. 2012. |
| 1153 | Spatial distribution of pCO2, $\Delta$ O2/Ar and dimethylsulfide (DMS) in polynya waters and |
| 1154 | the sea ice zone of the Amundsen Sea, Antarctica. Deep Sea Res. Part II Top. Stud.           |
| 1155 | Oceanogr. 71–76: 77–93. doi:10.1016/j.dsr2.2012.03.010                                       |

| 1156 | Tortell, P. D., C. Payne, C. Gueguen, R. F. Strzepek, P. W. Boyd, and B. Rost. 2008. Inorganic   |
|------|--------------------------------------------------------------------------------------------------|
| 1157 | carbon uptake by Southern Ocean phytoplankton. Limnol. Oceanogr. 53: 1266–1278.                  |
| 1158 | doi:10.4319/lo.2008.53.4.1266                                                                    |
| 1159 | Twining, B. S., and S. B. Baines. 2013. The trace metal composition of marine phytoplankton.     |
| 1160 | Annu. Rev. Mar. Sci. 5: 191–215. doi:10.1146/annurev-marine-121211-172322                        |
| 1161 | Valerio, A. M., M. Kampel, N. D. Ward, H. O. Sawakuchi, A. C. Cunha, and J. E. Richey. 2021.     |
| 1162 | CO2 partial pressure and fluxes in the Amazon River plume using in situ and remote               |
| 1163 | sensing data. Cont. Shelf Res. 215: 104348. doi:10.1016/j.csr.2021.104348                        |
| 1164 | Wuttig, K. and others. 2019. Critical evaluation of a seaFAST system for the analysis of trace   |
| 1165 | metals in marine samples. Talanta. doi:10.1016/j.talanta.2019.01.047                             |
| 1166 | Wyatt, N. J., A. Milne, E. M. S. Woodward, A. P. Rees, T. J. Browning, H. A. Bouman, P. J.       |
| 1167 | Worsfold, and M. C. Lohan. 2014. Biogeochemical cycling of dissolved zinc along the              |
| 1168 | GEOTRACES South Atlantic transect GA10 at 40°S. Glob. Biogeochem. Cycles 28: 44-                 |
| 1169 | 56. doi:10.1002/2013GB004637                                                                     |
| 1170 | Xu, Y., D. Shi, L. Aristilde, and F. M. M. Morel. 2012. The effect of pH on the uptake of zinc   |
| 1171 | and cadmium in marine phytoplankton: Possible role of weak complexes. Limnol.                    |
| 1172 | Oceanogr. doi:10.4319/lo.2012.57.1.0293                                                          |
| 1173 | Ye, N. and others. 2022. The role of zinc in the adaptive evolution of polar phytoplankton. Nat. |
| 1174 | Ecol. Evol. <b>6</b> : 965–978. doi:10.1038/s41559-022-01750-x                                   |
| 1175 | Zhang, Y., Z. Wen, M. P. Washburn, and L. Florens. 2010. Refinements to Label Free Proteome      |
| 1176 | Quantitation: How to Deal with Peptides Shared by Multiple Proteins. Anal. Chem. 82:             |
| 1177 | 2272–2281. doi:10.1021/ac9023999                                                                 |
| 1178 |                                                                                                  |