# Peer review of "Zinc stimulation of phytoplankton in a low carbon dioxide, coastal Antarctic environment"

_EGUsphere, 2025_

## Author Comment (AC1)

**Response to Reviews 7/2/25 (Responses in blue)**

**Zinc stimulation of phytoplankton in a low carbon dioxide, coastal Antarctic environment: evidence for the Zn hypothesis**

**RC1: 'Comment on egusphere-2025-1609', Anonymous Referee #1, 25 Apr 2025**

Kell et al., report an incubation based study to test whether primary producers in an Antarctic coastal environment respond to increased Zn availability. Whereas light, Fe and to a lesser extent Mn, are well established as drivers of productivity in Antarctic coastal ecosystems, any effects of Zn have not been well explored. The authors use multiple lines of argument to show that a state of co-limitation by Fe and Zn is possible. I am a trace metal chemist so cannot comment in depth on the metaproteomic or metatranscriptomic analyses. Overall I think the subject is topical and the text provides some interesting insights into Zn dynamics.

Minor comments (by line number):

I have not read much about dZn concentrations around Antarctica, I assume because it has not been measured much, if some values are reported in the literature I would find it interesting to refer to them in a few sentences just to understand what sort of range and normal profile should be expected in these coastal environments.

> Thank you for this comment, which allowed us to realize that additional context regarding dZn around Antarctica is needed in the introduction. There are actually several studies that have documented the distribution of dZn around Antarctica, and these measurements typically show nutrient-like vertical profiles. An example from Sieber et al 2020 showing typical nutrient-like profiles is linked in the paper below (see their Figure 2), in addition to our prior study showing nutrient-like profiles of dZn in the study region (Kell et al., 2024)

Sieber, M., Conway, T.M., de Souza, G.F., Hassler, C.S., Ellwood, M.J. and Vance, D., 2020. Cycling of zinc and its isotopes across multiple zones of the Southern Ocean: Insights from the Antarctic Circumnavigation Expedition. *Geochimica et Cosmochimica Acta*, *268*, pp.310-324.

Kell, R.M., Chmiel, R.J., Rao, D., Moran, D.M., McIlvin, M.R., Horner, T.J., Schanke, N.L., Sugiyama, I., Dunbar, R.B., DiTullio, G.R. and Saito, M.A., 2024. High metabolic zinc demand within native Amundsen and Ross sea phytoplankton communities determined by stable isotope uptake rate measurements. *Biogeosciences*, *21*(24), pp.5685-5706.

> We propose to add the following text to the introduction to introduce the current knowledge of dZn measurements in the Southern Ocean:

"Vertical profiles of dZn in the Southern Ocean have been measured previously. Zn has not historically been considered as a limiting micronutrient in the Southern Ocean due to the upwelling of nutrient-rich waters that bring dZn to nanomolar concentrations only a couple hundred meters below the surface. Yet nutrient-like profiles of dZn are evident throughout this region, with surface depletion due to biological uptake decreasing this large inventory in the upper water column (Fitzwater et al. 2000; Coale et al. 2005; Baars and Croot 2011; Sieber et al. 2020; Kell et al. 2024).

Additionally, both model-based estimates (Roshan et al. 2018) and direct field measurements (Kell et al. 2024) of Zn uptake in this region have demonstrated a substantial biological demand for Zn in surface waters, leading to significant dZn drawdown. This is consistent with and genomic and laboratory studies indicating an elevated Zn demand in polar phytoplankton (Twining and Baines 2013; Ye et al. 2022).

42-43 The concept of Zn limitation mainly applies to low $pCO_2$ environments which arise in various coastal areas for different reasons, it is not clear to me how $pCO_2$ in these "low" $CO_2$ zones will respond to future climate change as this likely depends on shifts in productivity, upwelling and freshwater discharge in addition to a slow increase in atmospheric $pCO_2$, so I didn't find this framing of changes in global $CO_2$ to be particularly relevant to the main story. I would have been more interested to know why these low $pCO_2$ zones exist, but maybe even this is getting a little away from the main focus of the text and I think the text would be fine without it.

> Line 42-43 was: "This study definitively establishes that Zn limitation can occur in the modern oceans, opening up new possibility space in our understanding of nutrient regulation of NPP through geologic time, and we consider the future of oceanic Zn limitation in the face of climate change."

>In this study, biology was the driver of the observed decrease in $pCO_2$, rather than freshwater input from glacial and sea ice melt. Please see our comment below. Due to the connection between Zn limitation and C acquisition, we feel maintaining this description is valuable, and as described below, we will modify the text to clarify this.

50 I would refer instead to the later Browning and Moore work (2023) if referring mainly to secondary limitation

> Line 50 was "Yet there is increasing evidence that other micronutrients such as zinc (Zn), cobalt (Co), and vitamin B12 can also influence phytoplankton productivity, often as secondary limiting nutrients after N, P, or Fe are added (Moore et al. 2013)."
> Thank you—we will add the 2023 Browning and Moore reference to this sentence.

68 I would suggest avoiding the term 'prejudice' as this implies unreasonable deductions. Consider that incubations to assess trace metal (co)/serial limitation are generally limited by the number of bottles that can be incubated simultaneously, so inevitably experiments lean towards designs which focus on the most deficient element, which is usually Fe, and perhaps include some combination of Mn, Co and Zn. This isn't unreasonable, but yes I agree with the notion that it means that co- or serial limitation by trace metals other than Fe has probably been under-appreciated to date. Perhaps the authors could rephrase.

> Line 68 was "Whether due to the early negative results, the few positive findings, or a general prejudice against considering additional factors in controlling marine productivity, it is our experience that there is currently no broad community recognition that zinc limitation is a process that could affect primary productivity in any region of the oceans, leaving the original 'zinc hypothesis' unresolved (Morel et al. 1994)."

> We appreciate this sentiment. The choice of language was based on our experience in prior submissions at other journals where reviewers vociferously argued that zinc could not be limiting in nature, despite our multiple lines of evidence and additional blank analyses. We propose to change the text to "Whether due to the early negative results, the few positive findings, or the practical

constraints of co-limitation studies in the field that limit the number of micronutrients tested, it is our experience that there is currently no broad community recognition that zinc limitation is a process that could affect primary productivity in any region of the oceans, leaving the original 'zinc hypothesis' unresolved (Morel et al. 1994)."

105 Not sure what 'total dissolved Fe' is, would just 'dissolved Fe' (and 'dissolved Zn') throughout not be clearer?

> We refer to our metal data as "total dissolved" metals following GEOTRACES terminology (see the GEOTRACES cookbook, Section 3.2 Total dissolved (filtered) samples: https://geotracesold.sedoo.fr/images/Cookbook.pdf)

128 I assume N+N means nitrate plus nitrite? Maybe define at first use (apologies if I missed this)

> Thank you for catching this, as N+N was not explicitly defined previously.

>We propose to change the text to "Consistent with high macronutrient abundance in this region, surface macronutrient concentrations were partially depleted at the experimental site with 64%, 46%, and 29% decreases in nitrate+nitrite (N+N), phosphate (P), and silicate (Si), respectively, comparing 10 m and average deep water (200 – 1000 m) values (Fig. 1o)."

142 Apologies if my terminology is wrong - is there a possibility of independent co-limitation i.e. both Zn and Fe produce positive, independent responses in the same species/groups?

> Line 142 was "However, addition of Zn alone (+Zn) also resulted in significantly higher chl a content compared to the controls (p = 0.011), implying that a subset of the incubated phytoplankton population benefitted from the addition of Zn alone, without additional Fe, and may thus have been experiencing primary Zn limitation (Fig. 2a)."

> Your interpretation is correct: that we saw a significant response in chl a with +Zn alone, this suggests Type I (Independent) Co-limitation, which we referred to in this sentence as "primary" Zn limitation.

>For clarity, we will change the text to "However, addition of Zn alone (+Zn) also resulted in significantly higher chl a content compared to the controls (p = 0.011), implying that a subset of the incubated phytoplankton population benefited from the addition of Zn alone, without additional Fe. This observation is consistent with independent co-limitation (Saito et al. 2008) (Fig. 2a), where two nutrients (such as Fe and Zn) each independently limit different subpopulations or processes, and adding either nutrient alone yields a response."

280-286 Do lab culture metal:P ratios diverge from field ratios? If so a comparison to whatever natural Zn:P ratios are available would be more convincing.

> Lines 280-286 were "Particulate Zn:C ratios reported previously in Zn-limiting culture studies of the diatom *Thalassiosira pseudonana* (Sunda and Huntsman 2005) were converted to Zn:P ratios using the Redfield ratio (Redfield 1958) (Supplementary Table 5). We then compared these ratios and associated growth rates with particulate Zn:P measured within biomass collected at 10, 25, 50 and 100 m at the experimental site. At each of these surface depths, Zn:P measured at the experimental site was ~ 2E-4 mol:mol, which, in comparison to cultured diatom Zn:P ratios, fell within the range of

severely Zn-limited growth rates (Supplementary Figure 6), again demonstrating the propensity for Zn-limited growth in this region and corroborating the incubation results."

> Metal:P ratios reported by Sunda (Sunda 2012;https://doi.org/10.3389/fmicb.2012.00204REF) do align with field expectations in many cases on the low/limiting side. The reason we are using the Zn:P ratios from culture studies is because those studies were conducted under conditions of Zn limitation, allowing us to define a Zn-limiting threshold. Prior Zn:P ratios from the field have not yet been connected to Zn-limiting conditions in the field, so unfortunately aren't useful in this comparison.

327-330 Not sure I agree with the logic of the connection here. Yes atmospheric $pCO_2$ is rising, but what are the drivers of low $pCO_2$ in these coastal areas where $CO_2$ is low? If productivity or freshwater discharge in these regions increases (which is quite plausible in some of the low $pCO_2$ areas highlighted), this may well maintain these regions in a state of low $CO_2$ in the future even with increasing atmospheric $pCO_2$.

>Lines 327-330 were "We then compared this laboratory-determined Zn/C limitation threshold estimate to both the in situ 221 µatm $pCO_2$ measured at our field study site, and to the historical, global trend in surface ocean $pCO_2$ (Fig. 4a,b). Global surface ocean $pCO_2$ levels are rapidly rising above both the laboratory-estimated 259 µatm $pCO_2$ Zn/C limitation threshold and our field observation value of 221 µatm (Jiang et al. 2023) (Fig. 4a,b). Though only a fraction of the modern-day surface ocean is currently at ≤ 250 ppm $pCO_2$ (predominantly comprised of polar regions; Fig. 4c), this represents a large decrease in oceanic extent compared to only 100 years ago (Fig. 4d)."

> To address this comment, we have calculated the impact of freshwater dilution on $pCO_2$ and found it to be small relative to biological uptake, as shown in the figures below.

> In this study, we documented a ~45% decrease in $pCO_2$ within Terra Nova Bay (~221 µatm) compared to values outside of TNB (~400µatm). Biology was the driver of this decrease in $pCO_2$, rather than freshwater input from glacial and sea ice melt. This is evident in the physicochemical data, where over the measured salinity range (S=33.6-34.8), the effect of simple dilution by fresh water input (DIC=Total Alkalinity=0) would result in a reduction of $pCO_2$ by only ~8-9 ppm (Plot A below). The signals we observe are much larger than that, consistent with a large phytoplankton uptake driver. The total alkalinity (TA) also does not change proportionally with DIC in this region (Plot B below), which is also not consistent with dilution driving a conservative mixing of TA and DIC.

[Figure]

(A)

[Figure]

(B)

Plots of pCO2 and Salinity (A) and TA and DIC (B) showing the range of variability observed during the CICLOPS expedition with Terra Nova Bay and sampling sites located outside of the bay showing properties consistent with biological CO2 drawdown rather than freshwater dilution.

>We will add this text to the manuscript draft to emphasize that the reduction of pCO2 within Terra Nova Bay was driven by biology rather than freshwater input: "In this study, we documented a ~45% decrease in pCO2 within Terra Nova Bay (~221 µatm) compared to values outside of TNB (~400µatm). Biology was the driver of this decrease in pCO2, rather than freshwater input from glacial and sea ice melt. This is evident in the physicochemical data, where over the measured salinity range (S=33.6-34.8), the effect of simple dilution by fresh water input (DIC=Total Alkalinity=0) would result in a reduction of pCO2 by only ~8-9 ppm. The signals we observe are much larger than that, consistent with a large phytoplankton uptake driver. The total alkalinity (TA) also does not change proportionally with DIC in this region, which is also not consistent with dilution driving a conservative mixing of TA and DIC."

> We agree that the future trajectory of $pCO_2$ in coastal regions will be influenced by a complex interplay of factors, including biological productivity and freshwater discharge. Our intention was not to pinpoint future $pCO_2$ trends in these regions, but rather to highlight that Zn status may be an important and underexplored factor influencing phytoplankton physiology and carbon cycling under low $pCO_2$ conditions. But it is true that our original text only discussed what would happen if pCO2 only increased globally (ie, less Zn limitation, maybe). The reviewer is making the good point that localized regions of low pCO2 regions could still persist.

> To caveat and clarify that our interpretation is not definitive, but rather intended to motivate further investigation into the role of Zn in coastal biogeochemistry, we will add this text to the discussion:

> "On the other hand, it is likely that despite rising $pCO_2$ levels, some coastal regions will continue to experience episodic or persistent low $pCO_2$ due to high productivity (as observed in this study), freshwater inputs, or other regional processes. Though we do not attempt to model future $pCO_2$ dynamics in these areas, our results suggest that Zn status may continue to be an important physiological constraint under low $pCO_2$ conditions, particularly in productive coastal systems. As such, Zn limitation should be considered as part of the broader framework for understanding carbon cycling in these regions, especially as they play a disproportionate role in global carbon export."

400 (and elsewhere in the methods), reference format is duplicated

> Thank you for catching this, we have corrected these duplicated references.

463-469 I assume the authors know this is not ideal, leaving samples unacidified for months usually lowers recovery, although having said that the effects of this on dZn appear to be not too bad, maybe add a comment (see Jensen et al., 2020, Assessment of the stability, sorption, and exchangeability of marine dissolved and colloidal metals)

>Lines 463-469 were "The analysis of total dissolved metals for this expedition has been described previously (Kell et al. 2024). Briefly, seawater collected shipboard by pressure-filtering X-Niskin bottles through an acid-washed 142 mm, 0.2 μM polyethersulfone Supor membrane filter (Pall) within 3 hours of rosette recovery using high purity (99.999%) N2 gas and stored at 4ºC. All sample collection occurred shipboard within an on-deck trace metal clean van. Samples were acidified to pH 1.7 with high purity HCl (Optima) within 7 months of collection and were stored acidified at room temperature for over 1 year prior to analysis."

> We appreciate that 7 months is a long time to wait prior to acidification, but this is short compared to the Jensen 2020 study (they stored samples unacidified for 22 months). We used a much longer acidification time (>1 year) compared to Jensen (5 months) to allow ample time for desorption from the polyethylene bottle walls. In addition, the Jansen study decanted their seawater samples to a new bottle prior to acidification, which loses all the wall-bound metals in the original bottle. Importantly, in this study we acidified our seawater in the original collection bottle to redissolve metals that had adsorbed to the walls.
>We will add the following text to this Methods paragraph: "This extended acidification time was used to counteract any loss of metal due to adsorption to the bottle walls (Jensen et al. 2020)."

---

## Author Comment (AC2)

**Response to Reviews 7/3/25 (Responses in blue)**

**Zinc stimulation of phytoplankton in a low carbon dioxide, coastal Antarctic environment: evidence for the Zn hypothesis**

RC2: 'Comment on egusphere-2025-1609', Anonymous Referee #2, 06 Jun 2025

This is an excellent study focused on teasing apart the complexities of phytoplankton nutrition, with a specific emphasis on overlooked and traditionally difficult-to-study trace metal nutrients in the environment, i.e., zinc. This study provides several lines of evidence that, in addition to iron (Fe), zinc (Zn) can also limit phytoplankton growth. Most of my comments are minor and largely relate to the accessibility and readability of the text.

>Thank you for your appreciation of the study.

To make the manuscript accessible to researchers outside marine environmental studies, it would be useful to define terms, such as "polynya".

> We will update the text using "polynya" to read: "A large phytoplankton bloom was present as indicated by high (> 3000 ng L-1) chlorophyll fluorescence concentrations in January that waned into February (Fig. 1d). This observation of high productivity is characteristic of Antarctic polynya environments, which are recurring regions of open water surrounded by sea ice (Arrigo et al. 2012)."

It would also help the reader when describing taxonomic classifications to give a little more information. An example, is line 89. For readers who are not experts in algal phylogeny to add "and the haptophyte Phaeocystis" or something similar, when first mentioning this alga. Is "Phaeocystis" used to refer to this alga (or algae) because the species is unknown or there are likely multiple species?

> Line 89 was "This phytoplankton community initially consisted of a mixed assemblage of both diatoms as indicated by fucoxanthin (fuco, Fig. 1e) and Phaeocystis as verified by shipboard microscopy and as indicated by 19'-hexanoyloxyfucoxanthin (19'-hex, Fig. 1f)."

> Phaeocystis antarctica (Phaeocystis hereon) is the dominant Phaeocystis species in this region (Arrigo et al., 1999 https://doi.org/10.1126/science.283.5400.365; DiTullio et al., 2003 https://doi.org/10.1029/078ARS03). It is true that all ZCRP hits for Phaeocystis (shown in Fig. 3) do indeed have P. antarctica as the best hit species.

> We will update the text to read: "This phytoplankton community initially consisted of a mixed assemblage of both diatoms as indicated by fucoxanthin (fuco, Fig. 1e) and the haptophyte *Phaeocystis* as verified…"

Sentences, such as "Pronounced and progressively deepening total dissolved Zn (dZnT) depletion over time was observed, with dZn depleted down to an average of 0.82 ± 0.47 nM at 10 m over all TNB stations (Fig. 1g)", are difficult to read and understand without re-reading. For instance, I had initial confusion about the use of "deepening" and "down". For instance, it appears "deepening" refers to depth in the water column, but the use of "down" refers to the dissolved Zn concentration decreasing? As this is a complex study, the authors need to be careful with word choice.

> For clarity we will reword this text to: "Additionally, we observed pronounced depletion of total dissolved Zn in surface waters across all TNB stations, with an average concentration of 0.82 ± 0.47 nM at 10 m (Fig. 1g). Notably, as the bloom progressed, this depletion extended progressively deeper into the water column (Fig. 1g), indicative of strong Zn uptake and export from the euphotic zone."

Since the first section of the results was largely previously published, this section could be shortened and/or moved to the methods section as a description of the sampled sites.

>We request to keep this brief section to provide environmental context without the reader having to read an additional manuscript.

Some clarity is needed with respect to whether the same station was measured temporally, or whether each date represents a different station.

>Each station (Table S2) was sampled only once, therefore each date represents a different station, but since all stations were spatially within TNB, we refer to this as temporal sampling of the region. >The text will be updated to: "Twenty-six stations within Terra Nova Bay (TNB) were temporally sampled over the course of one month (January 9 – February 18, 2018) during the 2017-2018 CICLOPS expedition (Fig. 1a; Supplementary Table 2) to concurrently characterize the natural progression of the phytoplankton bloom and biogeochemical changes in the water column (Kell et al. 2024). These stations were spatially distinct (each unique station was sampled once), but given that all were in relatively close proximity to each other within TNB (within a 52 km radius), we have combined the stations to create a temporal analysis of the region.

Do the authors have data to estimate the Zn content of the phytoplankton (or at least the plankton community) in sampled waters compared to how this value changes when the cells are fed Fe, Zn, and Fe+Zn? This would perhaps get at a better understanding of the community's Zn quota in relation to the amount of Zn that would be considered to be limiting, sufficient, or a luxury.

> This is a good suggestion, however, we did not save material from the experiments to conduct these measurements. Often material in these experiments is somewhat limited, time is short during experimental breakdown, and Zn in particular is notoriously difficult to collect particulate metal data from in the field due to the ease of contamination. We were not sure at the time that the Zn stable isotope uptake studies we conducted concurrently on this cruise would work, as our previous attempts had been contaminated, and prior to the accompanying study there were no field Zn stable isotope uptake data in the oceans. Hence, the experimental studies on laboratory-grown representative strains serves as a useful, and available, comparison.

ZCRP-A belongs to a very large and phylogenetically complex family. Algae, in particular, have an unusual number of paralogs from this family and many of these paralogs have distinct evolutionary origins. Are the authors confident that the "ZCRP-A" proteins identified are orthologs vs. potentially distinct paralogs of the characterized ZCRP-A? Similarly, the ZIPs are another complicated family, with some members expressed during Zn deficiency and others during Fe deficiency.

> We appreciate the reviewer's thoughtful comment regarding the complexity of the COG0523 and ZIP protein families and potential functional divergence among algal paralogs. Our identifications of ZCRP-A and ZIP protein hits (contigs) within the water column and incubation metaproteomic data

are based on high confidence BLAST hits to diatom reference ZCRP-A sequences (which we have characterized as related to Zn functionality; Kellogg et al., 2022 https://doi.org/10.1038/s41467-022-29603-y) and diatom reference ZIP1 protein sequences (Supp Table 4). This does not definitively distinguish orthologs from paralogs, so we have referred to them simply as "homologs". It is true that we are inferring Zn-related functionality to all metaproteomic contigs identified as ZCRPA or ZIP homologs, given the response to +Zn in the incubations and the scarce Zn concentrations in the surface water column in this study. To further increase our confidence that the metaproteomic contigs identified here have functionality with Zn (rather than other divalent metal cations), we entered each contig sequence into SHOOT (https://genomebiology.biomedcentral.com/articles/10.1186/s13059-022-02652-8) (https://shoot.bio/), which constructs a phylogenetic tree using the input sequence and identifies orthologs using the SHOOT database of organisms. Of the 21 unique contigs assigned as ZCRPA homologs, 19 were confirmed to be T.pseudonana orthologs, while 2 were assigned as orthologs to the Zn-related COG0523 E.coli proteins YjiA and YeiR. All 5 unique ZIP-assigned contigs were confirmed to be orthologs of T.pseudonana ZIP (Pfam PF02535). We propose to add this SHOOT analysis data of our identified ZCRPA and ZIP contigs as a new supplementary table.

---

## Author Response (AR2)

**Response to Reviews 7/29/25 (Responses in blue)**

**Zinc stimulation of phytoplankton in a low carbon dioxide, coastal Antarctic environment: evidence for the Zn hypothesis**

We thank the associate editor for their editorial comments, which we have addressed as follows:

Lines 42-43 This last phrase is not well connected to the rest of the sentence. Consider changing it to: '...and supporting the need to consider the future of oceanic Zn limitation in the face of climate change'

We have changed the previous sentence to this as requested.

In the same last sentence of the Abstract, 'definitively' is not needed, as the term 'establishes' already conveys that the result is firm.

We have removed "definitively".

Line 63 Delete '..' Removed '..' typo

Line 79 Delete 'it is our experience that' Removed this phrase

Line 181 Should be 'through 5 μm and 0.2 μm filters' Corrected this typo

Line 209 Should be 5 μm (and the > sign is not needed). Corrected this typo and removed > sign

Lines 324-325 Consider changing 'Antarctic waters are generally considered to not be prone to Zn limitation' to one of these alternatives:

'Antarctic waters are not generally considered to be prone to Zn limitation'

'Antarctic waters are generally considered not to be prone to Zn limitation'

We have changed the phrase to 'Antarctic waters are not generally considered to be prone to Zn limitation'

We thank the reviewers for their helpful comments in improving the manuscript. In addition to changes made to address the reviewers' comments, we have also made the following edits:

- Figure 1 now has station labels over panels (d) − (i).
- Supplementary Table 2 previously only listed stations in which the trace metal rosette (TMR) was deployed, and has now been updated to include all stations shown in Fig. 1.
- Our previous description of the protein extraction method as "bead-based" was incorrect—
  the correct "tube-gel" method is now described in detail in Methods. This does not affect our
  data nor conclusions. Additional details regarding the filters that biomass was collected on
  were also added.

RC1: 'Comment on egusphere-2025-1609', Anonymous Referee #1, 25 Apr 2025

Kell et al., report an incubation based study to test whether primary producers in an Antarctic coastal environment respond to increased Zn availability. Whereas light, Fe and to a lesser extent Mn, are well established as drivers of productivity in Antarctic coastal ecosystems, any effects of Zn have not been well explored. The authors use multiple lines of argument to show that a state of colimitation by Fe and Zn is possible. I am a trace metal chemist so cannot comment in depth on the metaproteomic or metatranscriptomic analyses. Overall I think the subject is topical and the text provides some interesting insights into Zn dynamics.

Minor comments (by line number):

I have not read much about dZn concentrations around Antarctica, I assume because it has not been measured much, if some values are reported in the literature I would find it interesting to refer to them in a few sentences just to understand what sort of range and normal profile should be expected in these coastal environments.

> Thank you for this comment, which allowed us to realize that additional context regarding dZn around Antarctica is needed in the introduction. There are actually several studies that have documented the distribution of dZn around Antarctica, and these measurements typically show nutrient-like vertical profiles. An example from Sieber et al 2020 showing typical nutrient-like profiles is linked in the paper below (see their Figure 2), in addition to our prior study showing nutrient-like profiles of dZn in the study region (Kell et al., 2024)

Sieber, M., Conway, T.M., de Souza, G.F., Hassler, C.S., Ellwood, M.J. and Vance, D., 2020. Cycling of zinc and its isotopes across multiple zones of the Southern Ocean: Insights from the Antarctic Circumnavigation Expedition. *Geochimica et Cosmochimica Acta*, 268, pp.310-324.

Kell, R.M., Chmiel, R.J., Rao, D., Moran, D.M., McIlvin, M.R., Horner, T.J., Schanke, N.L., Sugiyama, I., Dunbar, R.B., DiTullio, G.R. and Saito, M.A., 2024. High metabolic zinc demand within native Amundsen and Ross sea phytoplankton communities determined by stable isotope uptake rate measurements. *Biogeosciences*, *21*(24), pp.5685-5706.

> We have added the following text to the introduction to introduce the current knowledge of dZn measurements in the Southern Ocean:

"Vertical profiles of dZn in the Southern Ocean have been measured previously. Zn has not historically been considered as a limiting micronutrient in the Southern Ocean due to the upwelling of nutrient-rich waters that bring dZn to nanomolar concentrations only a couple hundred meters below the surface. Yet nutrient-like profiles of dZn are evident throughout this region, with surface depletion due to biological uptake decreasing this large inventory in the upper water column (Fitzwater et al. 2000; Coale et al. 2005; Baars and Croot 2011; Sieber et al. 2020; Kell et al. 2024). Additionally, both model-based estimates (Roshan et al. 2018) and direct field measurements (Kell et al. 2024) of Zn uptake in this region have demonstrated a substantial biological demand for Zn in surface waters, leading to significant dZn drawdown. This is consistent with and genomic and laboratory studies indicating an elevated Zn demand in polar phytoplankton (Twining and Baines 2013; Ye et al. 2022).

42-43 The concept of Zn limitation mainly applies to low pCO2 environments which arise in various

coastal areas for different reasons, it is not clear to me how pCO2 in these "low" CO2 zones will respond to future climate change as this likely depends on shifts in productivity, upwelling and freshwater discharge in addition to a slow increase in atmospheric pCO2, so I didn't find this framing of changes in global CO2 to be particularly relevant to the main story. I would have been more interested to know why these low pCO2 zones exist, but maybe even this is getting a little away from the main focus of the text and I think the text would be fine without it.

> Line 42-43 was: "This study definitively establishes that Zn limitation can occur in the modern oceans, opening up new possibility space in our understanding of nutrient regulation of NPP through geologic time, and we consider the future of oceanic Zn limitation in the face of climate change."

>In this study, biology was the driver of the observed decrease in pCO2, rather than freshwater input from glacial and sea ice melt. Please see our comment below. Due to the connection between Zn limitation and C acquisition, we feel maintaining this description is valuable, and as described below, we have modified the text to clarify this.

50 I would refer instead to the later Browning and Moore work (2023) if referring mainly to secondary limitation

- > Line 50 was "Yet there is increasing evidence that other micronutrients such as zinc (Zn), cobalt (Co), and vitamin B12 can also influence phytoplankton productivity, often as secondary limiting nutrients after N, P, or Fe are added (Moore et al. 2013)."
- > Thank you—we have added the 2023 Browning and Moore reference to this sentence.

68 I would suggest avoiding the term 'prejudice' as this implies unreasonable deductions. Consider that incubations to assess trace metal (co)/serial limitation are generally limited by the number of bottles that can be incubated simultaneously, so inevitably experiments lean towards designs which focus on the most deficient element, which is usually Fe, and perhaps include some combination of Mn, Co and Zn. This isn't unreasonable, but yes I agree with the notion that it means that co- or serial limitation by trace metals other than Fe has probably been under-appreciated to date. Perhaps the authors could rephrase.

- > Line 68 was "Whether due to the early negative results, the few positive findings, or a general prejudice against considering additional factors in controlling marine productivity, it is our experience that there is currently no broad community recognition that zinc limitation is a process that could affect primary productivity in any region of the oceans, leaving the original 'zinc hypothesis' unresolved (Morel et al. 1994)."
- > We appreciate this sentiment. The choice of language was based on our experience in prior submissions at other journals where reviewers vociferously argued that zinc could not be limiting in nature, despite our multiple lines of evidence and additional blank analyses. We have changed the text to "Whether due to the early negative results, the few positive findings, or the practical constraints of co-limitation studies in the field that limit the number of micronutrients tested, it is our experience that there is currently no broad community recognition that zinc limitation is a process that could affect primary productivity in any region of the oceans, leaving the original 'zinc hypothesis' unresolved (Morel et al. 1994)."

105 Not sure what 'total dissolved Fe' is, would just 'dissolved Fe' (and 'dissolved Zn') throughout not be clearer?

> We refer to our metal data as "total dissolved" metals following GEOTRACES terminology (see the GEOTRACES cookbook, Section 3.2 Total dissolved (filtered) samples: https://geotracesold.sedoo.fr/images/Cookbook.pdf)

128 I assume N+N means nitrate plus nitrite? Maybe define at first use (apologies if I missed this)

- > Thank you for catching this, as N+N was not explicitly defined previously.
- >We have changed the text to "Consistent with high macronutrient abundance in this region, surface macronutrient concentrations were partially depleted at the experimental site with 64%, 46%, and 29% decreases in nitrate+nitrite (N+N), phosphate (P), and silicate (Si), respectively, comparing 10 m and average deep water (200 1000 m) values (Fig. 1o)."
- 142 Apologies if my terminology is wrong is there a possibility of independent co-limitation i.e. both Zn and Fe produce positive, independent responses in the same species/groups?
- > Line 142 was "However, addition of Zn alone (+Zn) also resulted in significantly higher chl a content compared to the controls (p = 0.011), implying that a subset of the incubated phytoplankton population benefitted from the addition of Zn alone, without additional Fe, and may thus have been experiencing primary Zn limitation (Fig. 2a)."
- > Your interpretation is correct: that we saw a significant response in chl a with +Zn alone, this suggests Type I (Independent) Co-limitation, which we referred to in this sentence as "primary" Zn limitation.
- >For clarity, we have changed the text to "However, addition of Zn alone (+Zn) also resulted in significantly higher chl a content compared to the controls (p = 0.011), implying that a subset of the incubated phytoplankton population benefited from the addition of Zn alone, without additional Fe. This observation is consistent with independent co-limitation (Saito et al. 2008) (Fig. 2a), where two nutrients (such as Fe and Zn) each independently limit different subpopulations or processes, and adding either nutrient alone yields a response."
- 280-286 Do lab culture metal:P ratios diverge from field ratios? If so a comparison to whatever natural Zn:P ratios are available would be more convincing.
- > Lines 280-286 were "Particulate Zn:C ratios reported previously in Zn-limiting culture studies of the diatom *Thalassiosira pseudonana* (Sunda and Huntsman 2005) were converted to Zn:P ratios using the Redfield ratio (Redfield 1958) (Supplementary Table 5). We then compared these ratios and associated growth rates with particulate Zn:P measured within biomass collected at 10, 25, 50 and 100 m at the experimental site. At each of these surface depths, Zn:P measured at the experimental site was ~ 2E-4 mol:mol, which, in comparison to cultured diatom Zn:P ratios, fell within the range of severely Zn-limited growth rates (Supplementary Figure 6), again demonstrating the propensity for Zn-limited growth in this region and corroborating the incubation results."
- > Metal:P ratios reported by Sunda (Sunda 2012;https://doi.org/10.3389/fmicb.2012.00204REF) do align with field expectations in many cases on the low/limiting side. The reason we are using the Zn:P ratios from culture studies is because those studies were conducted under conditions of Zn limitation, allowing us to define a Zn-limiting threshold. Prior Zn:P ratios from the field have not yet

been connected to Zn-limiting conditions in the field, so unfortunately aren't useful in this comparison.

327-330 Not sure I agree with the logic of the connection here. Yes atmospheric pCO2 is rising, but what are the drivers of low pCO2 in these coastal areas where CO2 is low? If productivity or freshwater discharge in these regions increases (which is quite plausible in some of the low pCO2 areas highlighted), this may well maintain these regions in a state of low CO2 in the future even with increasing atmospheric pCO2.

>Lines 327-330 were "We then compared this laboratory-determined Zn/C limitation threshold estimate to both the in situ 221  $\mu$ atm pCO2 measured at our field study site, and to the historical, global trend in surface ocean pCO2 (Fig. 4a,b). Global surface ocean pCO2 levels are rapidly rising above both the laboratory-estimated 259  $\mu$ atm pCO2 Zn/C limitation threshold and our field observation value of 221  $\mu$ atm (Jiang et al. 2023) (Fig. 4a,b). Though only a fraction of the modern-day surface ocean is currently at  $\leq$  250 ppm pCO2 (predominantly comprised of polar regions; Fig. 4c), this represents a large decrease in oceanic extent compared to only 100 years ago (Fig. 4d)."

- > To address this comment, we have calculated the impact of freshwater dilution on pCO2 and found it to be small relative to biological uptake, as shown in the figures below.
- > In this study, we documented a ~45% decrease in pCO2 within Terra Nova Bay (~221  $\mu$ atm) compared to values outside of TNB (~400 $\mu$ atm). Biology was the driver of this decrease in pCO2, rather than freshwater input from glacial and sea ice melt. This is evident in the physicochemical data, where over the measured salinity range (S=33.6-34.8), the effect of simple dilution by fresh water input (DIC=Total Alkalinity=0) would result in a reduction of pCO2 by only ~8-9 ppm (Plot A below). The signals we observe are much larger than that, consistent with a large phytoplankton uptake driver. The total alkalinity (TA) also does not change proportionally with DIC in this region (Plot B below), which is also not consistent with dilution driving a conservative mixing of TA and DIC.

Plots of pCO2 and Salinity (A) and TA and DIC (B) showing the range of variability observed during the CICLOPS expedition with Terra Nova Bay and sampling sites located outside of the bay showing properties consistent with biological CO2 drawdown rather than freshwater dilution.

>We have added this text to the manuscript draft to emphasize that the reduction of pCO2 within Terra Nova Bay was driven by biology rather than freshwater input: "In this study, we documented a ~45% decrease in pCO2 within Terra Nova Bay (~221 µatm) compared to values outside of TNB (~400µatm). Biology was the driver of this decrease in pCO2, rather than freshwater input from glacial and sea ice melt. This is evident in the physicochemical data, where over the measured salinity range (S=33.6-34.8), the effect of simple dilution by fresh water input (DIC=Total Alkalinity=0) would result in a reduction of pCO2 by only ~8-9 ppm. The signals we observe are much larger than that, consistent with a large phytoplankton uptake driver. The total alkalinity (TA) also does not change proportionally with DIC in this region, which is also not consistent with dilution driving a conservative mixing of TA and DIC."

- > We agree that the future trajectory of pCO2 in coastal regions will be influenced by a complex interplay of factors, including biological productivity and freshwater discharge. Our intention was not to pinpoint future pCO2 trends in these regions, but rather to highlight that Zn status may be an important and underexplored factor influencing phytoplankton physiology and carbon cycling under low pCO2 conditions. But it is true that our original text only discussed what would happen if pCO2 only increased globally (ie, less Zn limitation, maybe). The reviewer is making the good point that localized regions of low pCO2 regions could still persist.
- > To caveat and clarify that our interpretation is not definitive, but rather intended to motivate further investigation into the role of Zn in coastal biogeochemistry, we have added this text to the discussion:
- > "On the other hand, it is likely that despite rising pCO2 levels, some coastal regions will continue to experience episodic or persistent low pCO2 due to high productivity (as observed in this study), freshwater inputs, or other regional processes. Though we do not attempt to model future pCO2 dynamics in these areas, our results suggest that Zn status may continue to be an important physiological constraint under low pCO2 conditions, particularly in productive coastal systems. As such, Zn limitation should be considered as part of the broader framework for understanding carbon cycling in these regions, especially as they play a disproportionate role in global carbon export."

400 (and elsewhere in the methods), reference format is duplicated

> Thank you for catching this, we have corrected these duplicated references.

463-469 I assume the authors know this is not ideal, leaving samples unacidified for months usually lowers recovery, although having said that the effects of this on dZn appear to be not too bad, maybe add a comment (see Jensen et al., 2020, Assessment of the stability, sorption, and exchangeability of marine dissolved and colloidal metals)

>Lines 463-469 were "The analysis of total dissolved metals for this expedition has been described previously (Kell et al. 2024). Briefly, seawater collected shipboard by pressure-filtering X-Niskin bottles through an acid-washed 142 mm, 0.2 μM polyethersulfone Supor membrane filter (Pall) within 3 hours of rosette recovery using high purity (99.999%) N2 gas and stored at 4°C. All sample collection occurred shipboard within an on-deck trace metal clean van. Samples were acidified to pH 1.7 with high purity HCl (Optima) within 7 months of collection and were stored acidified at room temperature for over 1 year prior to analysis."

> We appreciate that 7 months is a long time to wait prior to acidification, but this is short compared to the Jensen 2020 study (they stored samples unacidified for 22 months). We used a much longer acidification time (>1 year) compared to Jensen (5 months) to allow ample time for desorption from the polyethylene bottle walls. In addition, the Jansen study decanted their seawater samples to a new bottle prior to acidification, which loses all the wall-bound metals in the original bottle. Importantly, in this study we acidified our seawater in the original collection bottle to redissolve metals that had adsorbed to the walls.

>We have added the following text to this Methods paragraph: "This extended acidification time was used to counteract any loss of metal due to adsorption to the bottle walls (Jensen et al. 2020)."

**RC2: 'Comment on egusphere-2025-1609', Anonymous Referee #2, 06 Jun 2025**

This is an excellent study focused on teasing apart the complexities of phytoplankton nutrition, with a specific emphasis on overlooked and traditionally difficult-to-study trace metal nutrients in the environment, i.e., zinc. This study provides several lines of evidence that, in addition to iron (Fe), zinc (Zn) can also limit phytoplankton growth. Most of my comments are minor and largely relate to the accessibility and readability of the text.

>Thank you for your appreciation of the study.

To make the manuscript accessible to researchers outside marine environmental studies, it would be useful to define terms, such as "polynya".

> We have updated the text using "polynya" to read: "A large phytoplankton bloom was present as indicated by high (> 3000 ng L-1) chlorophyll fluorescence concentrations in January that waned into February (Fig. 1d). This observation of high productivity is characteristic of Antarctic polynya environments, which are recurring regions of open water surrounded by sea ice (Arrigo et al. 2012)."

It would also help the reader when describing taxonomic classifications to give a little more information. An example, is line 89. For readers who are not experts in algal phylogeny to add "and the haptophyte Phaeocystis" or something similar, when first mentioning this alga. Is "Phaeocystis" used to refer to this alga (or algae) because the species is unknown or there are likely multiple species?

- > Line 89 was "This phytoplankton community initially consisted of a mixed assemblage of both diatoms as indicated by fucoxanthin (fuco, Fig. 1e) and Phaeocystis as verified by shipboard microscopy and as indicated by 19'-hexanoyloxyfucoxanthin (19'-hex, Fig. 1f)."
- > Phaeocystis antarctica (Phaeocystis hereon) is the dominant Phaeocystis species in this region (Arrigo et al., 1999 https://doi.org/10.1126/science.283.5400.365; DiTullio et al., 2003 https://doi.org/10.1029/078ARS03). It is true that all ZCRP hits for Phaeocystis (shown in Fig. 3) do indeed have P. antarctica as the best hit species.
- > We have updated the text to read: "This phytoplankton community initially consisted of a mixed assemblage of both diatoms as indicated by fucoxanthin (fuco, Fig. 1e) and the haptophyte *Phaeocystis* as verified..."

Sentences, such as "Pronounced and progressively deepening total dissolved Zn (dZnT) depletion over time was observed, with dZn depleted down to an average of  $0.82 \pm 0.47$  nM at 10 m over all TNB stations (Fig. 1g)", are difficult to read and understand without re-reading. For instance, I had initial confusion about the use of "deepening" and "down". For instance, it appears "deepening" refers to depth in the water column, but the use of "down" refers to the dissolved Zn concentration decreasing? As this is a complex study, the authors need to be careful with word choice.

> For clarity we have reworded this text to: "Additionally, we observed pronounced depletion of total dissolved Zn in surface waters across all TNB stations, with an average concentration of 0.82 ± 0.47 nM at 10 m (Fig. 1g). Notably, as the bloom progressed, this depletion extended progressively deeper into the water column (Fig. 1g), indicative of strong Zn uptake and export from the euphotic zone."

Since the first section of the results was largely previously published, this section could be shortened and/or moved to the methods section as a description of the sampled sites.

>We request to keep this brief section to provide environmental context without the reader having to read an additional manuscript.

Some clarity is needed with respect to whether the same station was measured temporally, or whether each date represents a different station.

>Each station (Table S2) was sampled only once, therefore each date represents a different station, but since all stations were spatially within TNB, we refer to this as temporal sampling of the region. >The text will be updated to: "Twenty-six stations within Terra Nova Bay (TNB) were temporally sampled over the course of one month (January 9 – February 18, 2018) during the 2017-2018 CICLOPS expedition (Fig. 1a; Supplementary Table 2) to concurrently characterize the natural progression of the phytoplankton bloom and biogeochemical changes in the water column (Kell et al. 2024). These stations were spatially distinct (each unique station was sampled once), but given that all were in relatively close proximity to each other within TNB (within a 52 km radius), we have combined the stations to create a temporal analysis of the region.

Do the authors have data to estimate the Zn content of the phytoplankton (or at least the plankton community) in sampled waters compared to how this value changes when the cells are fed Fe, Zn, and Fe+Zn? This would perhaps get at a better understanding of the community's Zn quota in

relation to the amount of Zn that would be considered to be limiting, sufficient, or a luxury.

> This is a good suggestion, however, we did not save material from the experiments to conduct these measurements. Often material in these experiments is somewhat limited, time is short during experimental breakdown, and Zn in particular is notoriously difficult to collect particulate metal data from in the field due to the ease of contamination. We were not sure at the time that the Zn stable isotope uptake studies we conducted concurrently on this cruise would work, as our previous attempts had been contaminated, and prior to the accompanying study there were no field Zn stable isotope uptake data in the oceans. Hence, the experimental studies on laboratory-grown representative strains serves as a useful, and available, comparison.

ZCRP-A belongs to a very large and phylogenetically complex family. Algae, in particular, have an unusual number of paralogs from this family and many of these paralogs have distinct evolutionary origins. Are the authors confident that the "ZCRP-A" proteins identified are orthologs vs. potentially distinct paralogs of the characterized ZCRP-A? Similarly, the ZIPs are another complicated family, with some members expressed during Zn deficiency and others during Fe deficiency.

> We appreciate the reviewer's thoughtful comment regarding the complexity of the COG0523 and ZIP protein families and potential functional divergence among algal paralogs. Our identifications of ZCRP-A and ZIP protein hits (contigs) within the water column and incubation metaproteomic data are based on high confidence BLAST hits to diatom reference ZCRP-A sequences (which we have characterized as related to Zn functionality; Kellogg et al., 2022 https://doi.org/10.1038/s41467-022-29603-y) and diatom reference ZIP1 protein sequences (Supp Table 4). This does not definitively distinguish orthologs from paralogs, so we have referred to them simply as "homologs". It is true that we are inferring Zn-related functionality to all metaproteomic contigs identified as ZCRPA or ZIP homologs, given the response to +Zn in the incubations and the scarce Zn concentrations in the surface water column in this study. To further increase our confidence that the metaproteomic contigs identified here have functionality with Zn (rather than other divalent metal cations), we entered each contig sequence into SHOOT

(https://genomebiology.biomedcentral.com/articles/10.1186/s13059-022-02652-8) (https://shoot.bio/), which constructs a phylogenetic tree using the input sequence and identifies orthologs using the SHOOT database of organisms. Of the 21 unique contigs assigned as ZCRPA homologs, 19 were confirmed to be T.pseudonana orthologs, while 2 were assigned as orthologs to the Zn-related COG0523 E.coli proteins YjiA and YeiR. All 5 unique ZIP-assigned contigs were confirmed to be orthologs of T.pseudonana ZIP (Pfam PF02535). We have added this SHOOT analysis data of our identified ZCRPA and ZIP contigs as a new supplementary table (Supplementary Table 5).